# Super-shear ruptures steered by pre-stress heterogeneities during the 2023 Kahramanmaraş earthquake doublet

Kejie Chen [1,2,3] ✉, Guoguang Wei[1], Christopher Milliner [4], Luca Dal Zilio [5,6], Cunren Liang[7] & Jean-Philippe Avouac [4]

The 2023 M7.8 and M7.5 earthquake doublet near Kahramanmaraş, Turkey, provides insight regarding how large earthquakes rupture complex faults. Here we determine the faults geometry using surface ruptures and Synthetic Aperture Radar measurements, and the rupture kinematics from the joint inversion of high-rate Global Navigation Satellite System (GNSS), strong-motion waveforms, and GNSS static displacement. The M7.8 event initiated on a splay fault and subsequently propagated along the main East Anatolian Fault with an average rupture velocity between 3.0 and 4.0 km/s. In contrast, the M7.5 event demonstrated a bilateral supershear rupture of about 5.0–6.0 km/s over an 80 km length. Despite varying strike and dip angles, the sub-faults involved in the mainshock are nearly optimally oriented relative to the local stress tensor. The second event ruptured a fault misaligned with respect to the regional stress, also hinting at the effect of local stress heterogeneity in addition to a possible free surface effect.

As one of the most active intra-continental transform faults in the Eastern Mediterranean, the left-lateral East Anatolian Fault (EAF) has a history of destructive earthquakes[1]. This includes the very recent $M_w$ 6.7 Elazig earthquake in January 2020, which ended a period of relative quiescence which had followed a burst of earthquakes from 1871 to 1875[2]. The EAF has an intricate geometry with bends, step-overs, and sub-parallel faults, particularly in southern Turkey, where the EAF connects with the Dead Sea Fault (DSF) at the triple junction between the Arabian and African plates and the Anatolian block (Fig. 1). The segmented and complex geometry of the faults in that area might seem unfavorable for the development of large earthquakes. Nevertheless, on 6 February 2023 at 01:17:35 UTC, a M7.8 earthquake shook the southeastern parts of Turkey and northern Syria, followed ~9 h later by a M7.5 event along the Sürgü-Çardak (S-C) fault, situated 90 km from the initial M7.8 epicenter (https://tdvms.afad.gov.tr/ event_spec_data). This doublet shook the southeastern parts of Turkey and northern Syria resulting in >50,700 human casualties, marking it as the deadliest event in this region since the 525 Antioch earthquake[3]. The M7.8 event's epicenter is located on the Nurdagi-Pazarik Fault (NPF) splay fault, separate from the main EAF strand, implying a complex rupture history.

Most studies[4–6] suggest that the initial ~10 s of rupture occurred on the splay fault NPF of the main EAF strand. Slip on the EAF started only when the rupture reached their junction. The sub-sequent evolution of the rupture varies substantially among reports. Some argue for an instantaneous switch to a bilateral rupture[4,5], while back-projection imaging reveals a ~40–50 s lateral propagation toward the northeast before the southwestward bilateral rupture begins[6,7]. The rupture speed during the M7.8 event also remains a point of contention with potential evidence of a supershear rupture[8] while back-

[1]Department of Earth and Space Sciences, Southern University of Science and Technology, Shenzhen, China. [2]Guangdong Provincial Key Laboratory of Geophysical High-Resolution Imaging Technology, Southern University of Science and Technology, Shenzhen, China. [3]Institute of Risk Analysis, Prediction and Management (Risks-X), Academy for Advanced Interdisciplinary Studies, Southern University of Science and Technology, Shenzhen, China. [4]Division of Geological and Planetary Sciences, California Institute of Technology, Pasadena, CA, USA. [5]Earth Observatory of Singapore, Nanyang Technological University, Singapore, Singapore. [6]Asian School of the Environment, Nanyang Technological University, Singapore, Singapore. [7]School of Earth and Space Sciences, Peking University, Beijing, China. ✉e-mail: chenkj@sustech.edu.cn

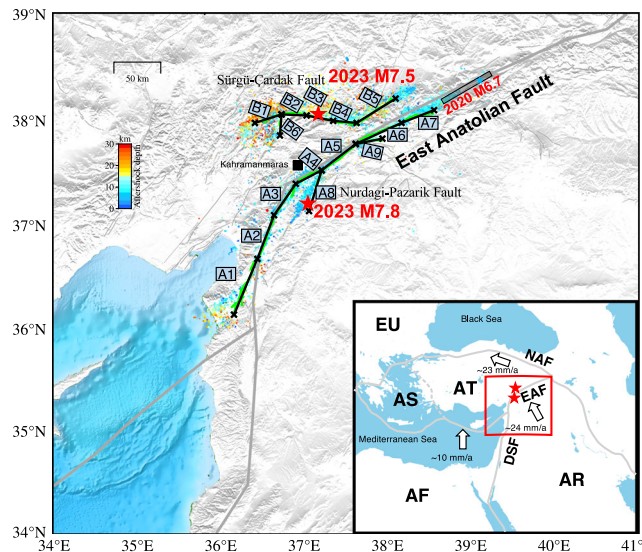

**Fig. 1 | Tectonic setting of the 2023 M7.8 and M7.5 Kahramanmaraş earthquake doublet.** Red stars indicate epicenters of the doublet. The blue star denotes the epicenter of the 2020 M6.7 event, the rupture extent of which is delineated with the black box from Chen et al.[2]. In this study the M7.8 rupture is divided into 9 planar sub-faults (A1–A9), and the M7.5 earthquake event, into 6 planar sub-faults (B1–B6). The green lines represent the surface rupture traces provided by Reitman et al.[54]. The locations of aftershocks for the first 20 days[26] are color-coded with depth. The inset map indicates major plate boundaries and plate velocities relative to Eurasia[55]. AF Africa, AR Arabia, AS Aegean Sea, AT Anatolia, EU Eurasia, DSF Dead Sea Fault, EAF East Anatolian Fault, NAF North Anatolian Fault.

projection results[9,10] and finite source modeling[4] suggest a sub-shear rupture speed between ~2 km/s and ~3.2 km/s. Regarding the M7.5 event, Jia et al.[9] found a supershear rupture velocity of 4.5 km/s, and Okuwaki et al.[11] proposed an even higher rupture speed of up to 6 km/s. These discrepancies emphasize the need for further exploration of this complex seismic event. Also, how such energetic and large-magnitude earthquakes could develop despite the rather complex fault geometry is puzzling. In this study, we investigate this question based on a detailed analysis of the rupture dynamics of the two earthquakes and the initial stress on the various fault segments that were activated during these events.

Here we undertake finite source inversions of the doublets using both dense near-field GNSS, and strong motion waveforms, constrained by remote sensing measurements. We first infer fault dips via Bayesian inversion using interferometric synthetic aperture radar (InSAR) measurements while fault strikes are constrained based on the surface ruptures determined from optical image correlation. We next present our kinematic finite source model. We further estimate the background stress field to investigate the mechanisms which might have enabled energetic ruptures of such a complex fault system.

## Results and discussion
### Surface ruptures and co-seismic measurements
To constrain the fault geometry, we calculate the 2D horizontal deformation from C-band Sentinel-1 SAR images and Sentinel-2 optical measurements by precise co-registration and sub-pixel correlation[12] (Supplementary Note 1). We next determine the 3D surface deformation (see Fig. 2a) by inverting the optical and radar measurements of surface motion in a total of six independent look directions (four from Sentinel-1 radar and two from Sentinel-2 optical pixel offsets, see detailed information in Supplementary Note 2 and Table 1, Fig. 1)[13]. The result is validated by comparing with the displacements measured at the GNSS stations (Supplementary Fig. 2) and similar pixel tracking geodetic imaging by other studies (Supplementary Fig. 3), the root

mean square error is also presented in Supplementary Fig. 4. The fault slip vector is measured from the fault-parallel discontinuity of 3D surface displacements using swath profiles. These profiles are oriented perpendicular to the rupture and stacked over a 4.8 km along-strike distance (30 pixels) to help reduce the effect of noise. Because the vertical displacements are poorly constrained, we only use the horizontal displacements. For each profile swath, we project the surface displacement into the local fault-parallel and fault-normal directions to estimate the strike-slip and fault-normal components of horizontal slip across the rupture. In practice, we invert the projected fault-parallel surface displacement by fitting how it varies with distance from the fault using the sum of linear and error functions (using Eq. (1) and (2) in ref. 14). The uncertainty is estimated from the Jacobian of the residuals with respect to the model parameters that are used to calculate the model covariance matrix. The surface displacements, the fault surface trace, and the fault slip measurements are presented in Fig. 2. These measurements are used to constrain the fault trace and local strike and incorporated in the slip inversion as detailed below.

### Bayesian inversion of dip angle for each fault segment
Based on the surface ruptures from the 3D deformation, we approximate the fault systems with nine planar fault segments for the M7.8 earthquake, and for the M7.5 earthquake, with six segments (Fig. 1). To estimate the subsurface dip angles for each segment, we model each fault segment as a rectangular planar fault with uniform slip. We estimated the model parameters in the Bayesian framework, with applying the down-sampled line of sight (LOS) displacements of the coseismic deformation maps from the ALOS-2 interferograms on both ascending and descending tracks (Supplementary Figs. 6–8) and the coseismic GNSS offsets (see "Methods"). We estimate the marginal posterior probability distribution of each parameter using a parallel sequential Markov Chain Monte Carlo (MCMC) sampler[15] with 1000 Markov chains. As such, each parameter space is characterized by 1000 samples, with dip angle estimates and uncertainties represented by the median and standard deviation of the sample distribution (Fig. 3 and Supplementary Table 4). The median model of the parameter samples indicates segmented dip angles along strike (Supplementary Fig. 9), and the estimated dip angles generally align with aftershock locations (Supplementary Fig. 10). The model accounts well for the ALOS-2 LOS displacements and the GNSS offsets (Supplementary Fig. 11). To summarize, all segments of the M7.8 event are near vertical, while the M7.5 event displays a gentler dip angle, particularly for the branches further from the epicenter, B1, B5 and B6, which only dip around 50° and are consistent with the estimations from He et al.[16] through a grid search of geodetic data fitting. We also note that other studies have inferred B4 and B5 varying from 50°[17] to be sub-vertical[9,18] based on aftershock distributions, implying the subsurface dipping angle estimates of fault segments by aftershock distributions can be controversial.

We then determine kinematic finite source models of both the M7.8 and M7.5 events using a standard methodology outlined in Method.

### Characteristics of the M7.8 event
Considering the inconsistencies among previous studies regarding the transition between the initial rupture of the splay fault and the development of the rupture on the main fault strand, we test delays from 0 to 50 s between northeastward and southwestward rupture initiation. We also vary the allowed maximum rupture speeds from 2.0 to 5.0 km/s, with variable intervals (from 0.2 to 1.0 km/s) to determine the favorable values and ease the computation burden as well. Our best fitting models require a 10 s delay for bilateral propagation, consistent with Jia et al.[9] as also constrained by strong motion waveforms.

The optimal rupture speeds are quite fast, falling in the range between 3.0 and 4.0 km/s. However, it is difficult to assess whether supershear velocities (exceeding the shear velocity of 3.5 km/s) are

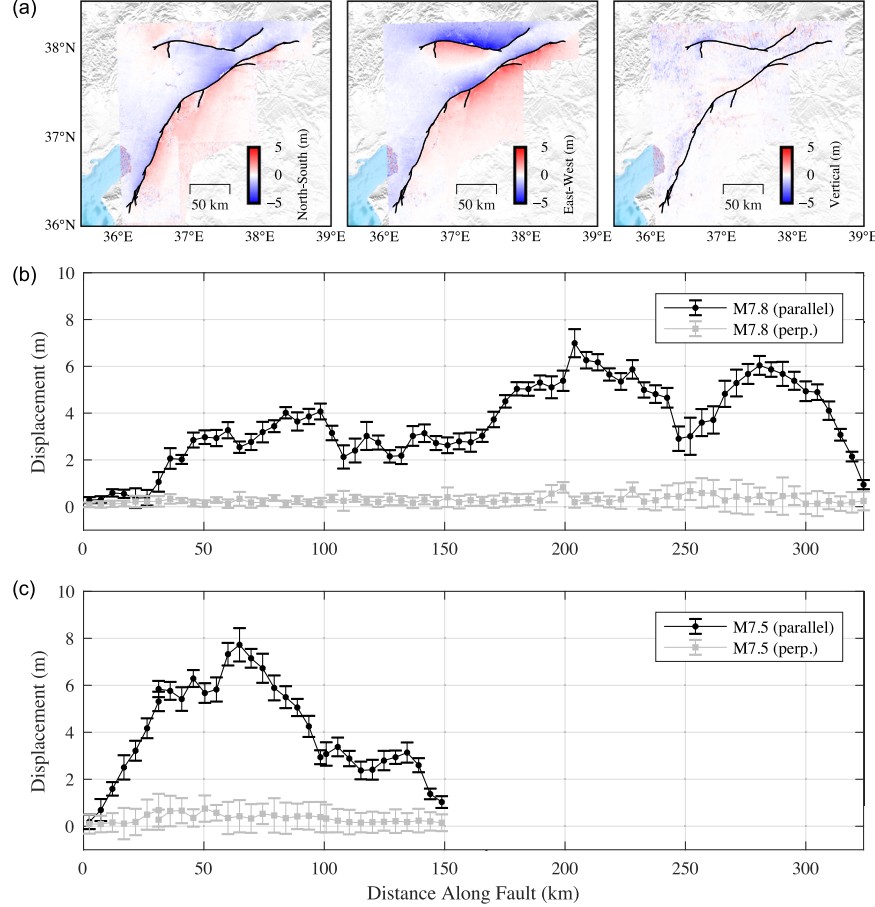

**Fig. 2 | 3D deformation field and surface rupture displacements. a** 3D deformation field derived from Sentinel-1 and Sentinel-2 image pixel offsets. From left to right are the north-south, east-west, and vertical components. Black lines denote the inferred surface rupture traces provided by Reitman et al. [54]. Surface rupture displacements, both horizontally perpendicular and parallel to the fault strike, along with their uncertainties, are calculated for the M7.8 event (**b**) and the M7.5 event (**c**). Distances along the fault are calculated from the westernmost point.

needed since the improvement in data fit from increasing the speed from 3.0 km/s to 4.0 km/s is marginal (see the data fits variance reductions against the rupture speeds in Supplementary Fig. 12). Besides, a finite source inversion through grid search would not resolve a transient supershear velocity.

Our reference slip model for the M7.8 event is shown in Fig. 4, while rupture propagation is illustrated in Fig. 5a and Supplementary Movie 1. Corresponding wave fits can be found in Supplementary Fig. 13 and sub-fault source time functions are present in Supplementary Fig. 14. The jackknife test (Supplementary Fig. 15) indicates that above 15 km depth, the slip can be well resolved. Our model indicates a moment of $7.76 \times 10^{20}$ Nm (Mw 7.86) released over 75 s, with peak moment rate at ~25 s. The nucleation splay fault NPF accounts for only a small portion of the total moment. The maximum slip is ~11 m, and the maximum slip rate reaches 3.0 m/s, located ~40 km from the epicenter. Although the bulk of the slip occurred above a 15 km depth, the slip depth of southwest segments varies significantly, with segment A2 notably shallower than neighboring segments A1 and A3. Dominant asperities primarily form around the junction and northeast segments.

### Characteristics of the M7.5 event

For the M7.5 event, finite source inversions and back-projections, generally agree on a supershear west of the epicenter and a subshear to the east[5,9,11]. However, these studies excluded the near-field GNSS stations (e.g., EKZ1, located just above the hypocenter), which provides crucial constraints in rupture speed inversion. Goldberg et al.[5] had to omit this station as it required an unrealistic >20 m maximum slip in their model. Conversely, we include both the static offsets and three-component waveforms at EKZ1, along with data from 15 other GNSS stations, in our inversion.

The rupture kinematics our reference model is depicted in Fig. 5b and Supplementary Movie 2, and slip distribution is shown in Fig. 6. The optimal velocity (see data fit variance reduction against rupture speed in Supplementary Fig. 16) reaches as high as 5.0–6.0 km/s, on segments B2, B3 and B4. It drops to 2.8 km/s on segments B1, B5 and B6. The model provides a satisfying fit to the GNSS and strong motion observations (Supplementary Fig. 17). Sub-fault source time functions are shown in Supplementary Fig. 18 and the total moment released is $5.57 \times 10^{20}$ Nm (equivalent to Mw 7.76), consistent with the solution from Goldberg et al. [5]. But the ~35 s duration is significantly shorter than that of the first M7.8 event, indicating a very energetic burst, also evidenced by a peak moment rate that is twice as large as that of the first shock. Maximum slip is similar ~11 m, but the bulk of the slip, especially on Segments B3 and B4, is deeper with acceptable uncertainty level (Supplementary Fig. 19). Notably, as the rupture propagates toward the northwest, the normal component becomes more prominent.

The bilateral supershear propagation for the initial ~40 km, contrasts with the previously suggested unilateral westward supershear propagation. This observation which challenges some existing models[9,18] but confirms others[17] is supported by the analysis incorporating data from the stations east side of the epicenter (e.g., GNSS stations EKZ1, MLY1), which shows a poor fit (Supplementary Fig. 20).

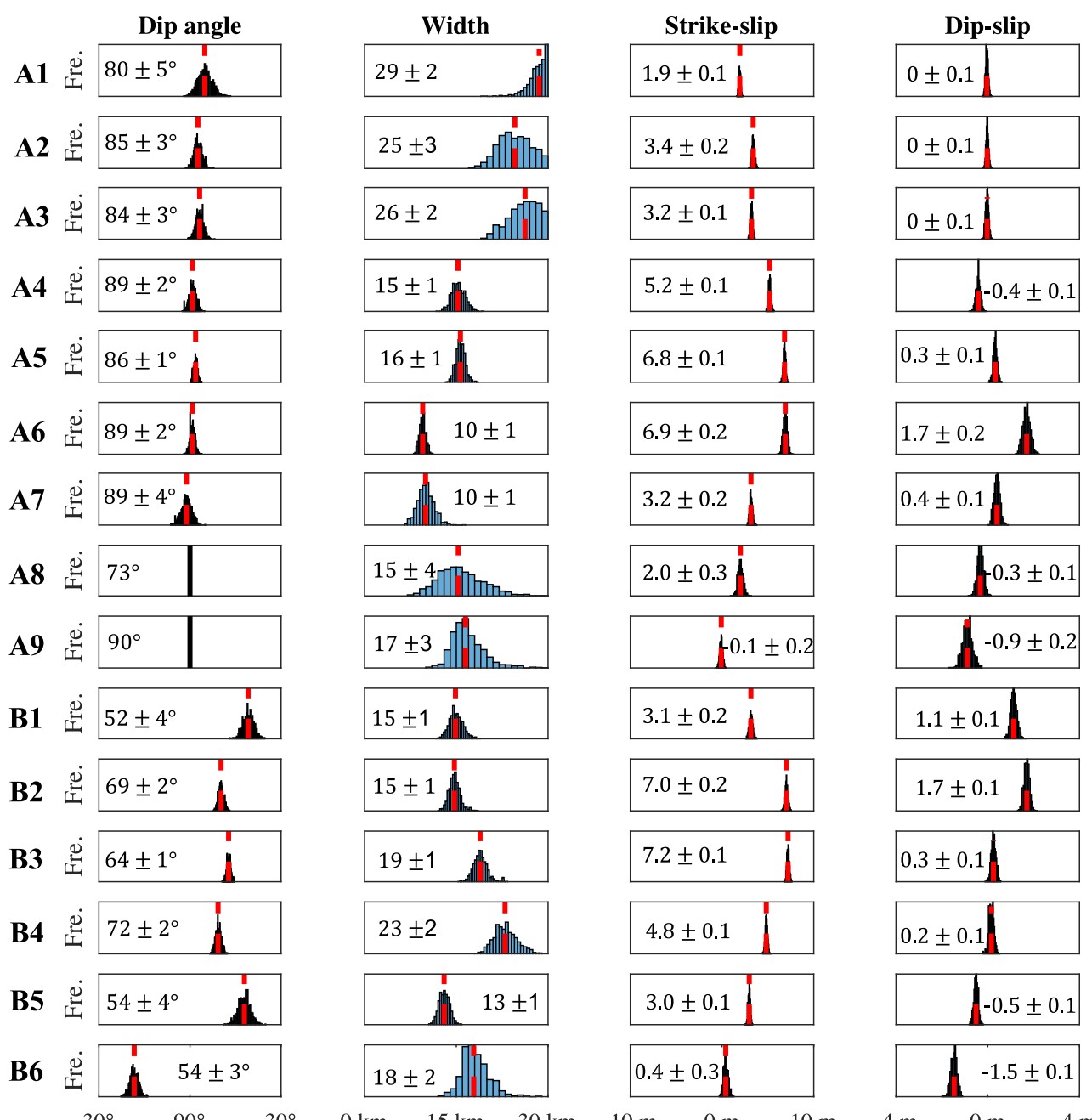

**Fig. 3 | Posterior samples of geometric parameters based on Bayesian inference.** The dashed red line marks the median of the samples for each parameter. The dip angle of segment A8 is set to be 73° to coincide with the hypocenter and the aftershock distribution, and segment A9 is set to be vertical at depth based on the aftershock distribution (Supplementary Fig. 8). In the inversion, the fault dip of the remaining segments is free. The fault plane of each segment can either dip to the right or the left of the segment. Dip angles samples within the range from 30° to 90° indicate dip angles for the assumed fault strike direction while sampling results within the range from 90° to 30° indicate dip angles for the opposite direction of the assumed fault strike.

Such dynamics underscore the benefit of incorporating a variety of observational data to capture the multifaceted nature of earthquake rupture processes.

### Effect of ambient pre-stress and dynamic stress along fault bends for the M7.8 rupture propagation

Sustained and fast rupture along the EAF-DSF intersection and southwest of it along a ~40° bend is a surprising feature of the M7.8 event as such a fault bend is typically thought to inhibit rupture propagation[19,20]. Understanding why rupture propagated past such a geometrical complexity and for another 175 km to the southwest is important for not only understanding the characteristics of the rupture but also more generally for understanding what factors limit the size of an earthquake. In theory, a rupture can branch on compressional or extensional bends at large angles away from the slipping crack as a result of dynamically increased shear stresses oriented away from the fault[21]. The favorable off-fault branch angle can be up to 70° and depends on the angle of the maximum compressive stress ($\sigma_1$) with the fault plane, the ratios in the pre-stress field, and the rupture velocity (with faster ruptures favoring larger bend angles). However, once initiation has occurred on the branching fault it must also be favorably oriented with the ambient stress field for rupture to continue and be sustained. We therefore analyze how the faulting kinematics compare to spatial variations of

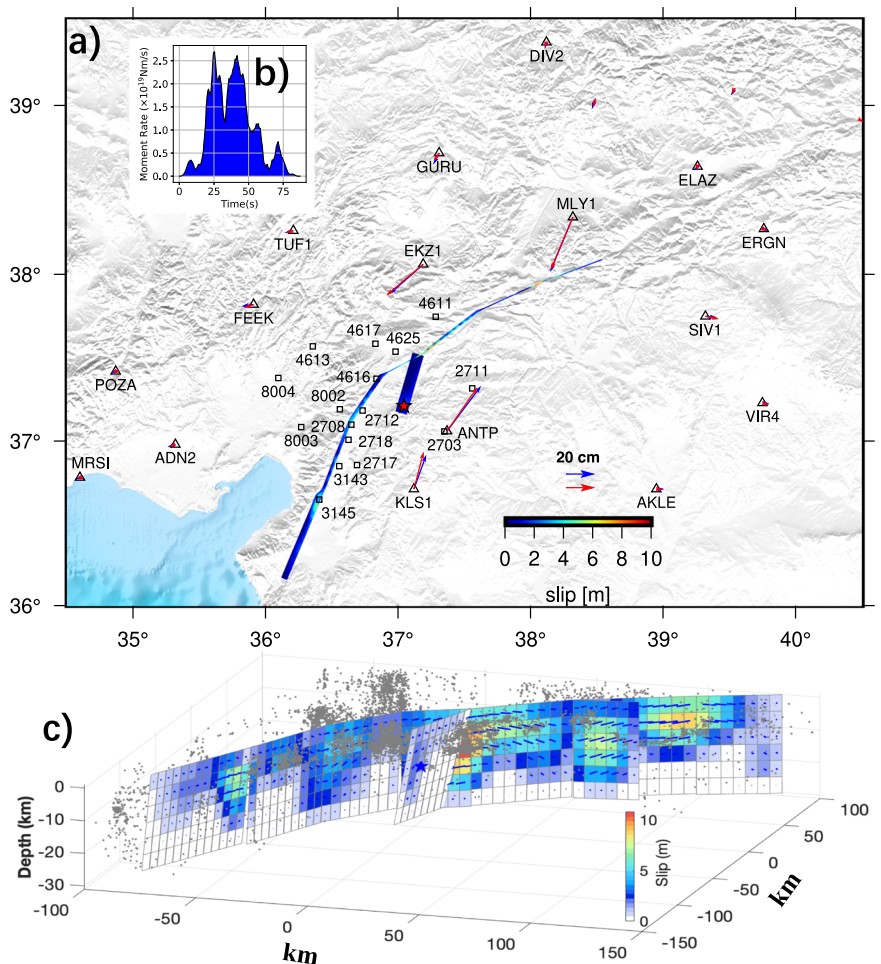

**Fig. 4 | Slip distribution and moment rate function for the M7.8 earthquake.**
**a** Map view of slip distribution for the M7.8 earthquake. The red star denotes the epicenter, while blue and red arrows show observed and synthesized GNSS offsets, respectively. Black squares locate the strong motion stations. **b** Moment rate function. **c** 3D view of the slip distributions with aftershocks from Ding et al. [26].

the pre-stress tensor derived from the focal mechanisms of local earthquakes.

To estimate the pre-stress tensor, we inverted 55 pre-event (2007–2020) focal mechanisms that were produced by Güvercin et al.[22] (see details in Supplementary Note 3 and Figs. 21–24), which clearly suggest a gradual variation of the orientation of the principal stresses. The method assumes that slip is parallel to the shear stress direction[23]. This gives a normalized deviatoric stress tensor in three zones located along the EAF (the southern zone covering segments A1–A3, the central zone covering segments A4-A5 and A8, and the northern zone including A6-A7 and A9). Qualitatively, our stress tensors agree with those from Güvercin et al.[22] which show a northeastward rotation of $\sigma_1$ along the M7.8 rupture from the Amanos segment in the southwest to the northeast. Note that we invert the stress field in four different zones based on the available focal mechanisms and our intent to assess how the stress field relates to the M7.8 and M7.5 ruptures.

From northeast to southwest along the M7.8 rupture, we find that the principal stresses show a ~ 20° anticlockwise rotation of $\sigma_1$ (Fig. 7) consistent with an Andersonian-type strike-slip stress regime in each stress zone. Plotting the orientation of the fault surfaces that ruptured in the M7.8 event with the pre-stress state on stereonets illustrates how almost all the fault surfaces were well aligned to the pre-stress field, as they are in regions of high fault instability ($I > 0.7$ in Fig. 7a–c). The fault instability quantifies how optimally aligned a fault surface is to the stress, with 0 being misaligned and 1 being most optimally aligned[24] (Fig. 7). This illustrates that even though there is a ~ 44° change in the

overall fault orientation along the entire length of the 320 km long rupture, from ~east-west striking in the northeast region of the M7.8 rupture (segments A6 and A7) to ~SW striking in the southwest (segments A1–A3), the fault surfaces are almost always optimally aligned because of the rotation of the ambient pre-stress field along-strike (where the fault instability is highest along the southernmost section with the mean value of fault patches of 0.98 and decreases slightly to 0.85 in the northeast). We also note that for each sub-fault, the mean slip vector (Supplementary Table 5) is closely parallel to the shear stress calculated based on this local stress tensor (Fig. 7).

Interestingly, we note that along a short segment of the rupture (segment A4, for ~20 km long) located southwest of where the Narli intersects the EAF, $\sigma_1$ is locally at a large angle to the EAF (of -73°). Our kinematic fault slip inversion also shows that the rupture velocity along this fault segment (A4) was fast (~3.8 km/s) and probably exceeded the Rayleigh speed (~3.5 km/s). Such conditions of a high rupture speed and locally highly oblique $\sigma_1$-fault angles are expected to produce dynamically increased shear and Coulomb stresses at larger branch fault angles of 40-90° on the extensional side of rupture (i.e., increasingly towards the southwest)[21]. This is an orientation that is consistent with the southwest trending bend of the EAF fault (located southwest of where the Narli intersects the EAF). Thus, under fault crack theory it seems that the rupture could have been sustained at high velocities along a large-scale continually bending fault and for a long distance due to a locally high angle of $\sigma_1$ which would have promoted high dynamic shear stresses at an angle away from the

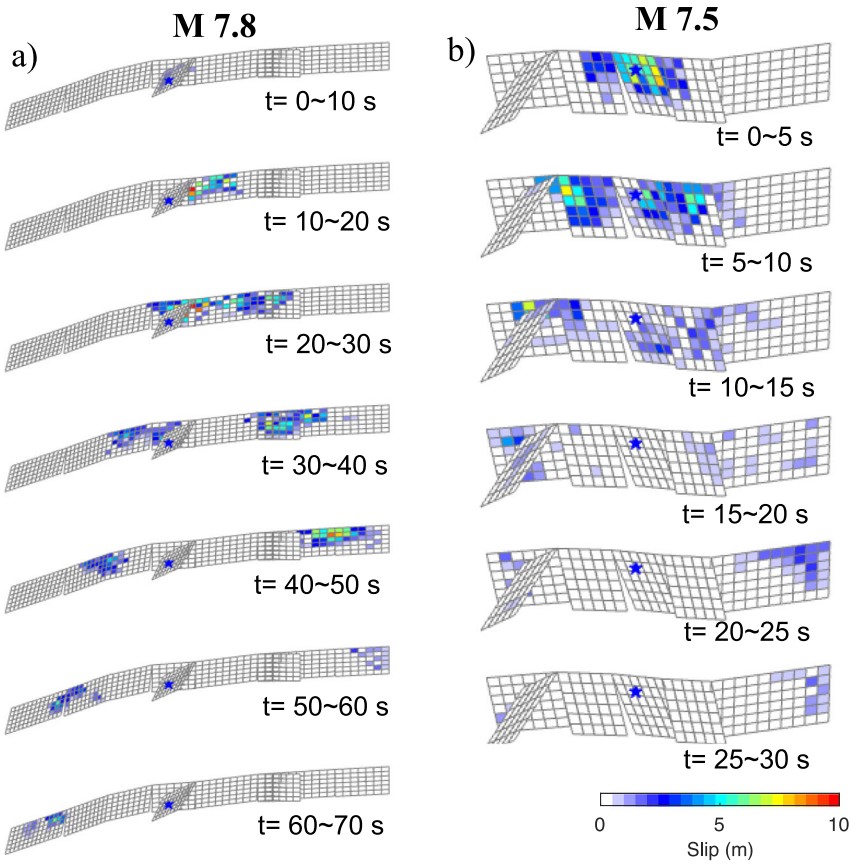

**Fig. 5 | Snapshot of rupture evolution. a** Snapshot of rupture evolution for the M7.8 event. **b** Snapshot of rupture evolution for the M7.5 event. The blue star denotes the epicenter.

propagating crack tip in the extensional slip direction (i.e., to the southwest) that coincides with the observed bend direction, and that due to the along-strike rotation of the ambient pre-stress, this would have allowed for rupture to be favorably oriented and sustained. We surmise that the dynamic simulations presented in refs. 9,25 are successful in reproducing the kinematics of the M7.8 event because the authors indeed assumed a northeastward rotation of $\sigma_1$ from the southwest to the northeast consistent with a nearly optimally oriented fault orientation along its curved geometry. Further dynamic simulations would however be needed to confirm that such an initial stress distribution is required to explain the extent of the observed rupture.

Furthermore, while most of fault traces of M7.8 events are optimally aligned with the pre-stress field, it is not entirely clear if the NPF is misaligned to the regional "central" pre-stress state or not as the fault strike of the NPF is not clear. Surface rupture observations (both field and pixel tracking) indicate it is -N19°E, which would be highly mis-orientated to the regional "central" stress state. However, relocated aftershocks from Ding et al.[26], indicate a number of more NE oriented structures where the hypocenter is located, indicating a strike of N38°E (Supplementary Fig. 25). Under the current "central" pre-stress state, the angle between $SH$ max and a fault strike (presuming it is -N38°E) would be 31.2, or a static friction of 0.61. However, if the NPF does have an orientation of N19°E, then this would suggest a spatially heterogeneous stress field would be required for it to be optimally aligned.

**The effect of spatial heterogeneity in pre-stresses for the M7.5 event**

The compact and large slip rupture that occurred on the S-C fault during the M7.5 earthquake implies a relatively large stress drop of the order of 11–21 MPa (calculated using *Knopoff*[27] formula for a strike-slip fault rupturing the free surface). The rather large stress drop and

supershear rupture velocity are surprising features given the faults are highly oblique to the main East Anatolian plate boundary fault, which suggests that they might be highly mis-orientated to the regional stress field. The fact that the rupture seems to have become supershear very early on is, in particular, surprising given the apparent misorientation to the regional stress field, as previous studies have suggested the early transition of ruptures to supershear requires a high pre-stress on the fault[28].

The M7.8 event imparted only a quite small static Coulomb and moderate dynamic stress changes imparted onto the majority of the length of the S-C faults (up to -1.5 MPa and 7 MPa, respectively, located only east of the hypocenter, with both being <1 MPa at and east of the hypocenter)[9]. A stress transfer from the first event cannot therefore explain why such a large and fast rupture occurred on an apparently misoriented fault. One possibility would be that the earthquake nucleated along an optimally oriented fault segment (B3) and that strong dynamic weakening would have allowed propagation of the rupture along the poorly aligned B5 and B6 segments. Such a scenario was proposed for the 2019 Mw 7.1 Ridgecrest event where rupture initiated on a small, optimally aligned fault segment that then propagated onto less-favorably orientated pre-existing faults[29].

Another possible explanation is that the local stress field prior to the 2023 earthquake sequence was locally significantly different from the regional stress field. The stress state obtained from inverting focal mechanisms from background seismicity[22] does show a moderate stress rotation along the S-C faults (Fig. 7d, with the most compressive stress ($\sigma_1$) orientated -N30°E) but indicating the faults would still be misaligned (with -60° angle between $\sigma_1$ and the S-C faults). However, it should be noted that the pre-2022 seismicity is very sparse with only 6 focal mechanisms along the S-C faults, which is not sufficient to robustly constrain the stress tensor[30]. The principal strain-rate field

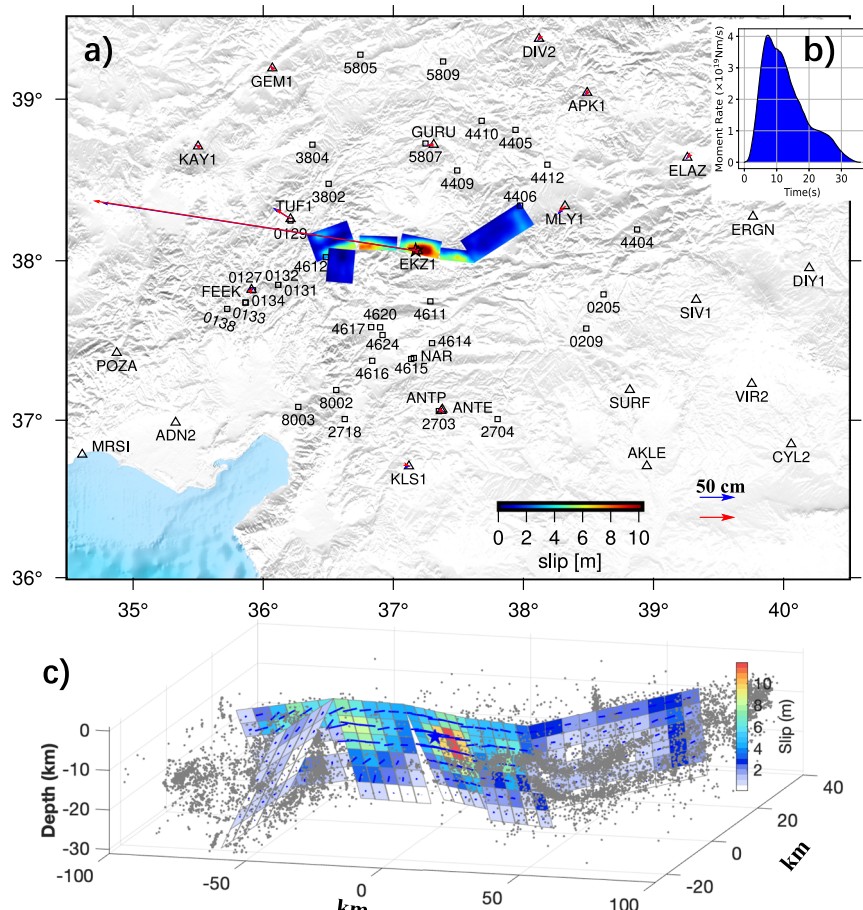

**Fig. 6 | Slip distribution and moment rate function for the M7.5 earthquake.**
**a** Map view of slip distribution for the M7.5 earthquake. The red star denotes the epicenter, while blue and red arrows show observed and synthesized GNSS offsets, respectively. Black squares locate the strong motion stations. **b** Moment rate function. **c** 3D view of the slip distributions with aftershocks from Ding et al.[26].

however, that is estimated from the surface velocity field using InSAR timeseries and GNSS data (from 2014-2019), shows a large, 30° rotation of the first principal strain-rate between the northeast region of the M7.8 rupture on the EAF (segments A6,7 and A9) to the region around the S-C[31]. Assuming the horizontal strain rate tensor is stationary and proportional to the horizontal stress tensor, which has been found to be valid for other areas such as Tibet and California[32,33], this would suggest $\sigma_1$ is at N47.5°E and only ~34° away from the S-C fault along most of its length (segments B2–B4). Under such a stress state most of the faults that ruptured during the M7.5 rupture would be well-orientated for failure (where fault locations are shown as white dots in Fig. 7e and located in regions of high fault instability of >0.75[24]. This would therefore not require a low initial static friction for failure to occur and can explain the prolonged rupture produced by physics-based dynamic rupture simulations[9]. We note that the north-south trending, west-dipping normal fault near the western end of the M7.5 termination has slip orientated in the opposite sense of that predicted under the stress regime (see Fig. 7d and e showing black vectors which are the observed slip direction with opposite motion sense to the gray vectors that are predicted from the stress regime). Normal slip on this structure is however consistent with the stress changes due to slip along the S-C fault as this subsidiary fault lies in an off-fault tensional stress lobe.

Thus, it seems significant spatial heterogeneity of the ambient pre-stress field over a short (a rotation of 30° over a 25 km distance) is likely present in this region, yet it is not clear what mechanism could cause such a large rotation of stresses. Possible reasons include a crustal weakness, such as a large major fault system that the S-C faults could present, or variations in the elastic material properties of the crust. The long and fast rupture that occurred during the M7.5 event on faults at a high oblique angle to the main plate boundary EAF system, which caused significant devastation to the local region, demonstrates the value in resolving local heterogeneity in stresses, so to understand whether major faults are misoriented faults or not and could fail with larger than expected ground shaking (e.g., see Fig. 5 of Jia et al.[9]). In regions lacking adequate background seismicity to estimate stress orientations, integrating the strain rate field could prove to be a useful alternative.

### Implications for rupture dynamics
The kinematics of the 2023 doublet earthquakes raise intriguing questions about the dynamics of rupture propagation on apparently non-optimally oriented strike-slip faults. It is interesting to investigate the processes or factors that enabled the rapid, ultimately supershear rupture propagation despite significant geometric complexities.

Our study shows that heterogeneities of pre-stress must have played a role to steer the two ruptures. We hypothesize that the dynamic simulations in refs. 9,25 are successful in reproducing the kinematics of the M7.8 event because the authors indeed assumed a northeastward rotation of $\sigma_1$ from the southwest to the northeast consistent with a nearly optimally oriented fault orientation along its curved geometry. However, further dynamic simulations are necessary to confirm whether this initial stress distribution is essential to explain the observed rupture extent.

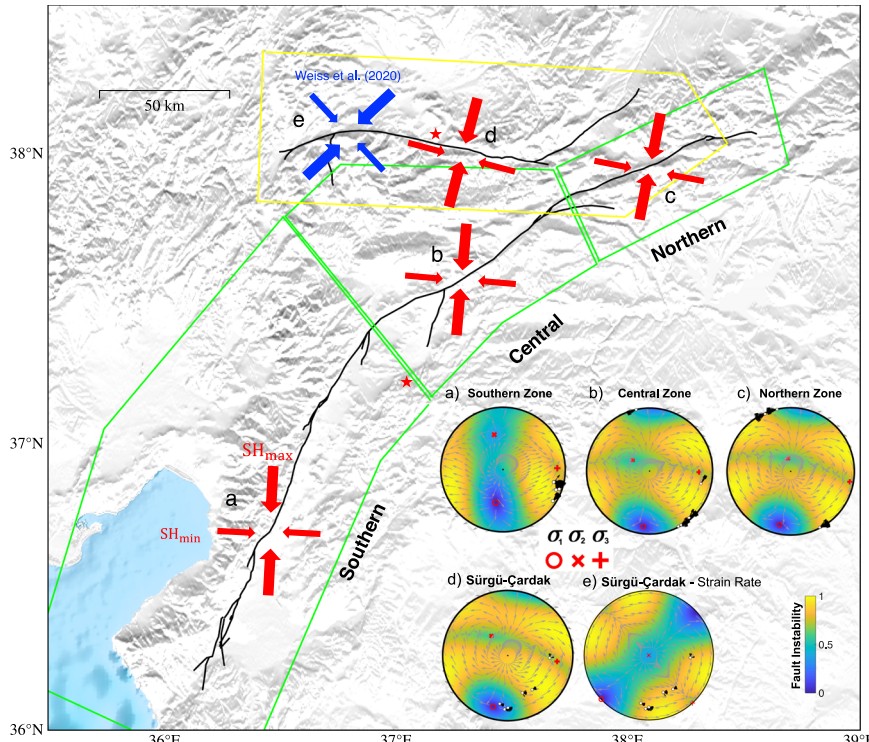

**Fig. 7 | Pre-stress tensors and fault-slip data for different zones.** Inset maps (**a**) show the southern zone (segments A1–A3), (**b**) show central zones (segments A4-A5 and A8), and (**c**) indicates northern zone (segments A6-A7 and A9). (**d**, **e**) are stress stereographs for the Sürgü-Çardak faults that ruptured during the M7.5 event, but (**d**) is derived from inverting focal mechanisms (similar to (**a**–**c**), and (**e**)) is derived from assuming the strain-rate tensor of Weiss et al.[31] is parallel to the stressing rate. Inset map blow-ups are also supplemented in Supplementary Fig. 26. Colors show the fault instability [*Vavrycuk*, 2013] which varies from 0 to 1 and characterizes how optimally aligned a fault is to the stress field (with 1 being most optimally aligned), calculated with a static friction of 0.6. Red symbols show the principal stress axes with labels referring to the stress stereograph panels. Gray vectors show the motion of the footwall expected under the pre-stress tensor. White dots show the pole-to-plane of all fault patches and black vectors show the footwall movement direction of each fault patch which are derived from the kinematic slip inversion. Thick black lines denote surface rupture traces from Reitman et al.[54], and green and yellow lines illustrate the extent of zones considered for stress inversion.

Nonetheless, fast ruptures of locally non-optimally oriented fault segments seem to have occurred on some of the fault segments ruptured in this earthquake sequence. The fast rupture along the EAF-DSF intersection, especially past a geometric bend of ~40°, is clearly anomalous. Commonly, such sharp geometric complexities are considered detrimental to continuous rupture propagation[34,35]. However, our observations support theoretical models indicating that dynamically increased shear stresses can sustain rupture along fault bends that deviate significantly from the primary crack[21,36]. These elastodynamic stresses can facilitate initiation on both compressional and extensional bends, governed by various factors such as the angle of primary stress with the fault plane. The observed kinematics, with faster ruptures crossing larger bend angles, align well with these studies.

The second event also exhibited fast ruptures on locally non-optimally oriented fault segments, producing strike-slip motion on several non-vertical, misaligned segments. Dipping faults, as explored in the studies by Oglesby et al.[37,38], introduce an interplay of the seismic waves with fault geometry and the Earth's surface. These effects have been primarily studied in the context of thrust faults where they result in a specific asymmetrical ground motion pattern, due to their shallow dip angle[37–39]. It is plausible that these mechanisms are relevant to non-vertical strike-slip faults, as observed in the M7.5 event. Along that line, Hu et al.[40] demonstrated that normal stress changes induced by rupture near the free surface in non-vertically dipping strike-slip faults can amplify slip and facilitate supershear rupture. The contrast of elastic properties across the fault could also be invoked. This bimaterial interface effect described by Andrews and Ben-Zion[41], can indeed help produce supershear ruptures. Finally, variation with depth of fault frictional properties could also have played a role. Kaneko and Lapusta[42] demonstrated that reduced strength at shallow depths could lead to local supershear transitions and investigated the specific pre-stress conditions necessary for these transitions. The concept seems applicable to the Kahramanmaraş earthquakes as well.

However, our study is not without limitations. Further exploration of the role and weighting of near-field stations in the inversion process is warranted. Additionally, despite efforts to account for uncertainties, the complexity and variability of geological structures imply that some degree of uncertainty inevitably remains. Future studies might aim to refine our understanding of these earthquakes and further test the models and methods applied in this study.

## Methods

### Co-seismic GNSS displacements retrieving

We use the GNSS co-seismic static deformation provided by Nevada Geodetic Laboratory (http://geodesy.unr.edu). To retrieve the 1-Hz GNSS co-seismic waveforms, we processed the raw data on 6 February 2023 collected from the Turkish Permanent GNSS Network (TUSAGA-Aktif) through epoch-wise precise point positioning[43], and detailed strategies could be found at Chen et al.[44]. Solid Earth tides, ocean tidal loading, and pole tides are accounted for following the IERS Conventions 2010[45]. Besides, to ensure a high signal-to-noise ratio, we only keep stations with >3 cm co-seismic static offsets and evident fluctuations in waveform time series. As a result, we finally incorporate static offsets at 26 stations and 14 stations for the M7.8 and M7.5 event source inversion, respectively. 1-Hz waveforms at 17 stations are used for both events, which are further trimmed to be 90 s long and filtered with a bandpass at a [0.005, 0.4] Hz corner frequency.

## Strong motion records

The earthquake doublet was well presented by the network of strong motion sensors run by Disaster And Emergency Management Presidency of Turkey (AFAD, http://tdvm.afad.gov.tr/). We selected 16 and 31 strong stations (see distribution in Figs. 4a and 5a) with good-quality observations for the M7.8 and M7.5 event source inversion. For each station, we downloaded its velocity waveforms directly from AFAD and decimated from 200 Hz to 1 Hz. The velocity waveforms were then filtered with a bandpass at [0.02,0.2] Hz and trimmed to be 90 s long, starting from the origin time of the earthquake.

## Processing of ALOS-2 measurements

We processed L-band ALOS-2 Scan SAR data, made freely available by the Japan Aerospace Exploration Agency (JAXA), post-earthquake, using alos2App.py in ISCE. This includes data from two tracks: ascending track 184 and descending track 77. The range split-spectrum method in alos2App.py mitigates strong ionospheric effects[46,47]. Following this, we unwrapped the InSAR phase using the Statistical-Cost, Network-Flow Algorithm for Phase Unwrapping (SNAPHU)[48] and geocode it to 3 arcsecond posting (~90 m) using the Shuttle Radar Topography Mission digital elevation model, see interferogram details in Supplementary Figs. 3 and 4. We masked out the rupture areas before phase unwrapping to avoid phase unwrapping errors.

Given the substantial pixel count in the ALOS-2 LOS deformation maps (Supplementary Fig. 5), we down-sampled the data using the quadtree method within the KITE software[49] (Supplementary Fig. 6), in which the covariance matrix of the down-sampled points is also estimated to account for data uncertainties. This process both mitigates data noise and renders the inversion more manageable.

## Bayesian inversion of fault geometry

We consider the statistical model of the observed data, $\mathbf{d} = \mathbf{G}(\theta)\mathbf{m} + \mathbf{e}$, where $\mathbf{d}$ is the vector representing the measured surface displacements by radar, and $\mathbf{e}$ is the vector representing the corresponding observation errors. The term $\mathbf{G}(\theta)\mathbf{m}$ represents the forward model, where $\theta$ represents the geometric parameters of the fault, and $\mathbf{G}(\theta)$ is the design matrix of Green's functions, which describes the surface displacements caused by unit fault slip in the elastic half-space. $\mathbf{m}$ represents the vector of fault slip parameters. For estimating subsurface fault dip angles of all segments, we modeled each fault segment as a rectangular fault plane with uniform slip. Therefore, for each fault segment, the geometric parameters include the dip angle of the fault plane and the along-strike length and along-dip width of the fault plane. The uniform slip of the fault plane is characterized by homogeneous strike-slip and dip-slip parameters. According to Bayes' theorem, the posterior probability distributions of the fault geometric parameters $\theta$ and slip parameters $\mathbf{m}$ depends on the prior probability distribution of the model parameters and the likelihood function formed by the difference between the observed data $\mathbf{d}$ and the forward model $\mathbf{G}(\theta)$[50]:

$$p(\theta, \mathbf{m}|\mathbf{d}) \propto p(\theta, \mathbf{m})exp\left\{-\frac{1}{2}[\mathbf{d} - \mathbf{G}(\theta)\mathbf{m}]^T\mathbf{C}_d^{-1}[\mathbf{d} - \mathbf{G}(\theta)\mathbf{m}]\right\} \quad (1)$$

where $p(\theta, \mathbf{m})$ represents the prior probability distributions of the fault geometric parameters $\theta$ and slip parameters $\mathbf{m}$. $\mathbf{C}_d$ represents the covariance matrix of the observed data. As segments A8 and A9 are two branches that did not cause obvious surface displacement gradient changes in the deformation maps (Fig. 2), the surface displacement measurements provide insignificant constraints on the subsurface fault dip angles of two segments. For simplicity, the dip angle of segment A8 is set to be 73° to coincide with the hypocenter and the aftershock distribution, and segment A9 is set to be vertical at depth based on the aftershock distribution (Supplementary Fig. 8), while the dip angles of the other segments are free within 30–90°. The

prior probability distributions of all model parameters to estimate are set with uniform distributions within given upper and lower bounds, which were summarized in Supplementary Table 2. To invert for these parameters, we used the down-sampled ALOS-2 LOS deformation as well as the GNSS offsets. We did not use the 3-D deformation maps because their accuracy was relatively poor compared with the ALOS-2 LOS deformation. Then the dip angle, the width along the dip, and the strike-slip and dip-slip components of all segments were simultaneously sampled using the sequential MCMC technique[15]. We summarized the standard deviation of posterior samples for all parameters to characterize the parameter uncertainties (Fig. 3 and Supplementary Table 4). Our preferred model was chosen based on the median of posterior samples (Supplementary Fig. 7), which demonstrated a good fit to the data (Supplementary Fig. 9).

## Finite source inversion

We determine the time evolution of fault slip from the joint inversion of high-rate GNSS, strong-motion waveforms, and GNSS static displacement by multi-time-window method[51]. The fault segments have their geometry imposed based on the result of the Bayesian inversion and are further subdivided into ~5 × 5 km² subfaults. Along dip, the number of subfaults is fixed at six, while along the strike, the number varies based on the surface rupture length. In total, we have 468 and 222 subfaults for the M7.8 and M7.5 earthquakes, respectively. We allow for rake variations with ±25° of the GCMT solution for the M7.8 event. However, for the M7.5 event, due to a significant normal faulting component, we set the two slip vectors to be −80° and 10°. The source time function is represented by five overlapping symmetric triangles, each with a duration of 5 seconds and overlapping by 2.5 s.

When determining rupture speeds, we fully consider published back-projection results and use grid search to determine optimal values for different sections. For the M7.8 event, we assume that the NPF, as well as segments to the northeast and the southwest of the junction, each have their individual rupture speeds. As a result, the rupture velocities are divided the into three groups: Group I for segment A8, Group II for segments A5, A6, A7, and A9, and Group III for segments A1 to A4. For the M7.5 event, considering that sharp changes in fault orientation could halt supershear[52], we divided velocities into two groups: Group I comprises segments B2, B3, and B4, while Group II includes B1, B5, and B6. To ensure the stability of the inversion results, we employed the first-order Laplacian regularization method[51].

Green's functions for dynamic waveforms and static offsets are calculated based on the layered velocity structure proposed by Güvercin et al. [22] (Supplementary Table 3) using the frequency-wavenumber integration method developed by Zhu and Rivera[53]. We also apply the same bandpass filter used for the waveforms to the Green's functions. For data weighting, each data type is normalized by its own norm. After testing different weighting factors, we find that for the two events, equal weighting fits all data reasonably well. Furthermore, to evaluate the reliability and uncertainty of the inversion results, we performed a jackknife test by randomly excluding 20% of the data.

## Data availability

GNSS data used in this study were obtained from the Turkish Permanent GNSS Network (TUSAGA-Aktif) and are available at https://www.tusaga-aktif.gov.tr/Web/DepremVerileri.aspx. Strong motion records were provided by AFAD (https://tdvms.afad.gov.tr/list-station/543428/37.043/37.288, https://tdvms.afad.gov.tr/list-station/543593/37.239/38.089). ALOS-2 data are made freely available by the Japan Aerospace Exploration Agency (JAXA) at: https://www.eorc.jaxa.jp/ALOS/en/dataset/alos_open_and_free_e.htm. Sentinel-1 data are also freely available and provided by the European Space Agency (ESA) under the Copernicus Program.

## Code availability

Finite source inversion code is modified based on Mudpy (https://github.com/dmelgarm/MudPy), and the calculation of Green's function for Bayesian inversion is cutde (https://github.com/tbenthompson/cutde).

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

## Acknowledgements

K.C. was funded by the National Natural Science Foundation of China (NSFC) 42074024 and Guangdong Provincial Key Laboratory of Geophysical High-resolution Imaging Technology (2022B1212010002). C.M. was supported by the NASA Earth Surface and Interior Focus Area ROSES grants (80NM0018D0004, 80NSSC20K0492). L.D.Z was supported by the European Research Council (ERC) Synergy Grant "Fault Activation and Earthquake Rupture" (FEAR) (No. 856559), the Earth Observatory of Singapore (EOS), and the Singapore Ministry of Education Tier 3b project "Investigating Volcano and Earthquake Science and Technology (InVEST)" (Award No. MOE-MOET32021-0002). C.L. was supported by NSFC 42274026 and J.P.A was supported by NASA/ROSES grant 80NSSC20K0492. We thank JAXA for providing ALOS-2 data used in this study.

## Author contributions

K.C. carried out finite source modeling and drafted the initial manuscript; G.W. performed the Bayesian inversion of fault geometries; C.M. processed optical offset data and inverted the background stress; C.L. processed the InSAR measurements; L.D.Z and J.P.A. helped with the data analysis and the writing. All authors took part in finalizing the paper.

## Competing interests

The authors declare no competing interests.
