## [Peer Review File · Nature Communications]

Super-shear ruptures steered by pre-stress heterogeneities during the 2023 Kahramanmaraş earthquake doubletEditorial Note: Parts of this Peer Review File have been redacted as indicated to remove third-party material where no permission to publish could be obtained.

Reviewers' comments:

Reviewer #1 (Remarks to the Author):

This is a very interesting paper that uses observational data to constrain the fault geometry of the 2023 Kahramanmaraş earthquake doublet, and then performs slip inversions to solve for the time history of rupture and slip in these events. Most importantly, the authors then make estimates of the heterogeneous stress field in the vicinity of the causative faults, and discuss their kinematic results in this context. They thereby find plausible physical mechanisms for many of their observations, including how rotation of the stress field allows rapid rupture along the earlier fault with quite variable strike, and the ability to propagate along an apparently misoriented fault system in the later event. They come to the very important conclusion that a characterization of the heterogeneity of the local stress field may be crucial in estimating seismic hazard; a simple large-scale, regional stress field may lead to erroneous results. The scientific questions are laid out clearly, and the methods are explained well. The results of their kinematic inverse modeling are largely consistent with prior work, which lends plausibility to their discussion of the potential dynamic sources for these results. Of course, it would be fascinating if the authors had tested some of their dynamic hypotheses with 3D dynamic modeling, but I can see that this might be beyond the scope of the paper (and I suspect it will be the subject of a follow-up paper from this group).

I was particularly interested in the authors' discussion of how the shallow dip of the second event might have facilitated supershear rupture in this case. They note that the work of Oglesby et al. (1998, 2000) indicates that interactions between the radiated stress field and the free surface could conceivably lead to an enhanced likelihood of free-surface-induced supershear rupture (e.g., Kaneko and Lapusta, 2010). The authors don't give any detailed description of how this interaction might come about, though. I believe that the authors are correct in their hunch here, and I may be able to suggest a specific mechanism. Hu et al. [2020] modeled the free-surface-induced supershear transition for non-vertically dipping strike-slip faults, and found that normal stress changes induced by rupture near the free surface can amplify slip and facilitate supershear rupture in one direction of propagation. I suggest that this effect could explain the unilateral supershear propagation to the west of the hypocenter and its absence to the east. Figure 8, panels b and c in Hu et al. [2020] would correspond to eastward and westward propagation, respectively (b would actually be the mirror image of the eastward propagation, with identical dynamics, because eastward propagation with left-lateral slip is equivalent to westward propagation with right-lateral slip). There is clearly more supershear rupture in panel c (westward) compared to panel b (eastward), and it is quite plausible that with different parameterizations, there could be no supershear in panel b while retaining supershear in panel c. The reason for this effect is that the different direction of rupture propagation induces normal stress increments of opposite sign, somewhat analogous to the bimaterial interface [Andrews and Ben-Zion, 1997] and even more analogous to the effects seen with asymmetric topography [Kyriakopoulos et al., 2021] (See figure 9 in Hu et al. [2020]). For reasons explained in Kyriakopoulos et al. [2021], propagation to the west produces a second-order free surface interaction analogous to a thrust fault, with an amplification

of the dynamic stress drop near the surface, while propagation to the east is more analogous to a normal fault, with de-amplification of the dynamic stress drop near the surface. The authors are welcome to use this interpretation in their work, but I certainly won't demand it; they should feel free to ignore or disagree with my suggestion.

I have very little to criticize about this manuscript. At first I didn't fully understand what was new and novel about this paper that would set it apart from prior kinematic/inverse models of this earthquake sequence, but after reading the entire manuscript I can see that the interpretation in light of the inferred heterogeneous stress field is truly novel. I suggest that they bring this point out even more strongly in the abstract so that the reader will know what to look for in the paper. Some minor suggestions follow, along with the pertinent locations in the manuscript.

Figure 2b: What direction does "perpendicular" refer to? Horizontal and perpendicular to strike, or vertical?

208-220: I'm a little confused by the rotation angles in this section. The fault changes its orientation by around 44 degrees as one progresses along strike, and the stress changes its orientation by around 20 degrees over the same length. That means that there must be a ~20 degree range in the angle of the stress with respect to the local fault orientation between different portions of the fault. Do the authors still consider all these segments to be close to optimal orientation, even with such a strong variation in the angle of the stress with respect to the fault?

276-277: Could the authors clarify a little more how they explain the slip on the normal fault being in the direction opposite that which would be expected from the local regional stress field? Are they saying that this small normal fault is largely loaded by the nearby strike-slip fault, not by the regional tectonics?

In conclusion, I think this is an excellent paper that will be of great interest to the community. I think it will be suitable for publication with very minor revisions.

Andrews, D. J., and Y. Ben-Zion (1997), Wrinkle-like slip pulse on a fault between different materials, *Journal of Geophysical Research*, 102(1), 552-571.

Hu, F., D. D. Oglesby, and X. Chen (2020), The Near-Fault Ground Motion Characteristics of Sustained and Unsustained Free Surface-Induced Supershear Rupture on Strike-Slip Faults, *Journal of Geophysical Research: Solid Earth*, 125(5), doi:10.1029/2019JB019039.

Kyriakopoulos, C., B. Wu, and D. D. Oglesby (2021), Asymmetric topography causes normal stress perturbations at the rupture front: The case of the Cajon pass, *Geophysical Research Letters*, 48(20), e2021GL095397.

Reviewer #2 (Remarks to the Author):

The study obtained geodetic observations, and then performed slip model inversion and stress analysis for the 2023 Turkey earthquake doublet. The authors concluded that the fault geometry matches with the stress tensor derived from focal mechanisms, and argued that stress heterogeneity and free surface effect have an impact on the complex rupture process of the earthquake. However, there are key issues in the geodetic data processing, finite fault inversions and stress analysis, which have to be addressed before the manuscript could be considered for a publication.

The authors presented insufficient details for their geodetic data processing, i.e., on the unwrapped interferograms and offset images. They also failed providing a self-consistent description about their 3D deformations and fault geometry/slip inversion. Although they derived 3D deformations, they did not perform validation on the results, which show obvious orbital errors. The authors derived 3D deformations but they didn't use them in fault geometry/slip inversion. They only used SAR offset data to constrain the model. The model residuals also show obvious systematic error (the constant value). Near some of the fault segments, the down-sampled data are very sparse, hence the resulted uncertainties are larger than the other segments. The authors should perform more data uncertainty analysis as well as their impact on the model derived from them.

For stress inversion, the authors only provided the final stress directions but no uncertainties, which could be misleading and difficult to interpret. It is not clear how can the authors derive robust stress fields using only 55 focal mechanisms without uncertainty analysis.

In finite fault inversions, the authors did not rigorously separate the deformation of one event from another before using them in the inversion. The waveform fitting quality is also not so good, in particular compared with those already published.

Major Comments:

#1 The authors claimed they processed all ALOS-2 ScanSAR data, but they provided no details in supplemental materials. The readers could be misled or confused if no wrapped or unwrapped interferograms are presented.

#2 Section "Remote Sensing" is also misleading. The title includes optimal images, but I cannot find any introduction about Sentinel-2 data processing in the main text. There is only brief writeup in the supplement, but not connected with the main text. This should be further improved.

#3 The authors mentioned they used the range split-spectrum method to mitigate ionospheric noise. This is confusing because range split spectrum's accuracy is quite sensitive to unwrapping errors. If the fringes are very dense, SNAPHU may fail, and range split-spectrum will lead to unavoidable errors in interferograms. The final fringes can be much denser or dramatically sparser if the estimated ionospheric noise includes unwrapping errors. For these two earthquakes, the unwrapping operation is very challenging. Please see the unwrapping operation provided by Xu et.al in their Communications Earth and Environment preprint (<https://github.com/yunjunz/2023-Turkey-EQ>).

#4 The authors should give more details on phase unwrapping. There is no unwrapped interferograms in the manuscript and supplement materials, but clear near-fault observations were shown in Figure S2 and S4. The results of the InSAR observations in Fig.S2 and S4 should not be InSAR data, instead, they should be SAR range offset images.

#5 The north-south deformation in Fig.2a contains obvious stripe errors. That is due to offset

images from Sentinel-2 are contaminated by orbital error. The authors only produced InSAR interferograms and some Sentinel-1 range offsets. They are all in the range direction which is not sensitive to the north-south direction. Therefore, only the north-south offset image from Sentinel-2 and the azimuthal offset from Sentinel-1 (these results should be very noisy although the authors didn't provide these azimuth offsets) contributes most to the final north-south deformation. Its orbital errors are all inherited to the final deformation. I strongly suggest the authors to perform Multiple Aperture Interferometry on ScanSAR data, which can improve the data quality in particularly for north-south deformation.

#6 The authors should compare their 3D deformations to 3D GNSS offsets to validate the accuracy of 3D deformation.

#7 The down-sampled deformation results (near Amanos fault) given in Fig.S2 and Fig.S4 are very sparse. The inverted dip angle should have large uncertainty here. The authors only give the MCMC uncertainty on their slip model, but they didn't perform any resolution or uncertainty tests for their slip model. These two tests are quite important to clarify if the data can resolve fault slip and/or geometry inversion robustly.

#8 Following #5, the aftershock catalog near A1 (Fig.1) shows shallow dip angle (at least smaller than 70 degree) but the inversion showed a result of ~80 degree. The authors should double check their data and the MCMC process. A resolution test would be needed as well.

#9 The authors also present no detail on the stress inversion. More details should be given to verify if 55 pre-event (2007-2020) focal mechanisms can derive robust principal stress directions. Please give sufficient details on how to derive the stress tensor and the uncertainty, as well as on the diversity of slip vectors.

#10 Figure S3 is not properly presented, with no figure caption. The strong motion station IDs could not be seen in the figure. The waveform fitting quality in Fig.S3 and Fig.S5 are not so good.

#11 The authors simply truncated the geodetic deformation of one event from another (Fig. S2 and S4) in space, without considering the actually impact of the deformation from one event to another, this is not appropriate.

Reviewer #3 (Remarks to the Author):

Chen et al., utilized abundant geodetic and seismic data to investigate the rupture kinematics of the 2023 Kahramanmaras, Turkey M7.8 earthquake doublets. They propose that sub-faults involved in the M7.8 and M7.5 events align closely to optimal orientations of derived stress tensor from historical earthquake focal mechanisms. I think it's an interesting and consistent scientific insight to explain the initiation and propagation of large earthquake events, but I have several major and minor comments before the paper is accepted to publish.

Major comments:

1. Lines 219-220 and fig 7: Can you please show the strike angle of each principal stress axis and the mean slip vector? It's clearer to have a table that compares the fault strike angle and corresponding optimal orientation of each local stress tensor, assuming a static friction of 0.6 in your case.

2. Fig 7 shows that most of fault traces of M7.8 events are optimally aligned with the pre-stress field, however, the NPF where the M7.8 earthquake initiated is not aligned with the “central” pre-stress field. Can you further explain why the mainshock initiated along a splay fault that doesn’t have the optimal crack direction?

3. Your lines 246-247 state that the early transition to regional supershear rupture requires a high pre-stress on the fault. The evidence from Weiss et al., 2020 suggests a 30-degree rotation of the first principal stress tensor (assuming strain rate and stress have the same tensor). I am curious how such a stress axis rotation would allow for high initial shear stress quantitatively?

4. Lines 251-252: In the case of 2019 M7.1 Ridgecrest earthquake, Jin and Fialko, 2020 reveals a scenario that the mainshock rupture can nucleate on a small but more optimally oriented fault but propagated along less-favorably oriented pre-existing faults subsequently due to dynamic weakening.

Minor comments:

1. Line 34: Please add the fault trace of 2020 Jan M6.7 Elazig earthquake into fig 1, because it’s also occurred along the East Anatolian Fault (EAF).

2. Line 174: Since the geodetic moment magnitude calculated from your slip model is 7.8, why did you assume the moment magnitude of the second earthquake is 7.5 not 7.8?

3. Lines 149 & 176-177: The second event has a similar moment magnitude (M7.8) but a concentrated slip distribution compared to the first one. Why does your slip model reveal the peak slip of both earthquakes is ~ 10 meters? According to Jia et al., 2023 and Liu et al., 2023, both their slip models reveal that the peak slip is even larger than 12 m.

4. Line 658: In figure 2a, why doesn’t the fault trace of M7.5 event match with the polarity boundary of north-south displacements?

5. Figure 3: Why are dip angles of A8 and A9 fixed to 73 and 90 degrees, respectively? Are these two dip angles constrained precisely by the relocated seismicity?

6. Figure 7: Where are the gray vectors? The red symbols that represent the principal stress axes and white dots are too small to see in those inset maps.

Reviewers' comments:

Reviewer #1 (Remarks to the Author):

This is a very interesting paper that uses observational data to constrain the fault geometry of the 2023 Kahramanmaras
earthquake doublet, and then performs slip inversions to solve for the time history of rupture and slip in these events. Most
importantly, the authors then make estimates of the heterogeneous stress field in the vicinity of the causative faults, and
discuss their kinematic results in this context. They thereby find plausible physical mechanisms for many of their
observations, including how rotation of the stress field allows rapid rupture along the earlier fault with quite variable strike,
and the ability to propagate along an apparently misoriented fault system in the later event. They come to the very important
conclusion that a characterization of the heterogeneity of the local stress field may be crucial in estimating seismic hazard; a
simple large-scale, regional stress field may lead to erroneous results. The scientific questions are laid out clearly, and the
methods are explained well. The results of their kinematic inverse modeling are largely consistent with prior work, which
lends plausibility to their discussion of the potential dynamic sources for these results. Of course, it would be fascinating if
the authors had tested some of their dynamic hypotheses with 3D dynamic modeling, but I can see that this might be beyond
the scope of the paper (and I suspect it will be the subject of a follow-up paper from this group).

I was particularly interested in the authors' discussion of how the shallow dip of the second event might have facilitated
supershear rupture in this case. They note that the work of Oglesby et al. (1998, 2000) indicates that interactions between the
radiated stress field and the free surface could conceivably lead to an enhanced likelihood of free-surface-induced supershear
rupture (e.g., Kaneko and Lapusta, 2010). The authors don't give any detailed description of how this interaction might come
about, though. I believe that the authors are correct in their hunch here, and I may be able to suggest a specific mechanism. Hu
et al. [2020] modeled the free-surface-induced supershear transition for non-vertically dipping strike-slip faults, and found that
normal stress changes induced by rupture near the free surface can amplify slip and facilitate supershear rupture in one direction
of propagation. I suggest that this effect could explain the unilateral supershear propagation to the west of the hypocenter and
its absence to the east. Figure 8, panels b and c in Hu et al. [2020] would correspond to eastward and westward propagation,
respectively (b would actually be the mirror image of the eastward propagation, with identical dynamics, because eastward
propagation with left-lateral slip is equivalent to westward propagation with right-lateral slip). There is clearly more supershear
rupture in panel c (westward) compared to panel b (eastward), and it is quite plausible that with different parameterizations,
there could be no supershear in panel b while retaining supershear in panel c. The reason for this effect is that the different
direction of rupture propagation induces normal stress increments of opposite sign, somewhat analogous to the bimaterial
interface [Andrews and Ben-Zion, 1997] and even more analogous to the effects seen with asymmetric topography
[Kyriakopoulos et al., 2021] (See figure 9 in Hu et al. [2020]). For reasons explained in Kyriakopoulos et al. [2021],
propagation to the west produces a second-order free surface interaction analogous to a thrust fault, with an amplification of
the dynamic stress drop near the surface, while propagation to the east is more analogous to a normal fault, with de-

amplification of the dynamic stress drop near the surface. The authors are welcome to use this interpretation in their work, but
I certainly won't demand it; they should feel free to ignore or disagree with my suggestion.

Response: We appreciate your insightful suggestion and the opportunity to further clarify the dynamics of the supershear
rupture observed in the M7.5 earthquake. The references you provided, including Hu et al. [2020], Andrews and Ben-Zion
[1997], and Kyriakopoulos et al. [2021], offer significant insights into the mechanisms of supershear rupture, especially in the
context of free-surface interactions and fault geometry. We agree that these references are complementary to those we had
selected, and we now quote them in the revision.

Upon revisiting our findings in light of your comment, we realized the necessity to clarify an important aspect of our results.
Contrary to the unilateral supershear propagation to the west that many studies have suggested, our analysis, supported by
stations eastside of the epicenter (e.g., GNSS stations EKZ1, MLY1), and corroborated by Liu et al. [2023, Nature
Communications], indicates a bilateral supershear rupture near the epicenter for the initial 40 km of the rupture process. This
observation suggests a more complex rupture dynamic than previously thought, with implications for understanding the stress
field interactions and the resultant rupture propagation directions.

Incorporating this clarification, we have revised the "Implications for Rupture Dynamics" section of our manuscript. We now
include a detailed discussion on how our observations of bilateral supershear rupture near the epicenter align with the
theoretical framework provided by Hu et al. [2020] and how this nuanced understanding impacts the interpretation of surface
interactions and rupture dynamics. Specifically, we address how the bilateral nature of the supershear rupture challenges and
extends the current models, including considerations for the normal stress changes induced by rupture near the free surface, as
well as the implications of asymmetric topography and bimaterial interface effects on supershear propagation.

Furthermore, we now emphasize better the poor fit of data at stations eastward of the epicenter obtained under the assumption
of unilateral westward supershear propagation. This observation highlights the benefit of integrating high-resolution data into
analyses to achieve a more accurate representation of the earthquake's dynamics.

We believe these revisions and clarifications provide a more comprehensive and accurate account of the rupture dynamics
observed during the M7.5 earthquake, offering valuable insights into the complex interplay between fault geometry, rupture
propagation, and surface interactions. We are grateful for your suggestions and believe that incorporating these insights has
significantly enriched our manuscript.

I have very little to criticize about this manuscript. At first I didn't fully understand what was new and novel about this paper
that would set it apart from prior kinematic/inverse models of this earthquake sequence, but after reading the entire

manuscript I can see that the interpretation in light of the inferred heterogenous stress field is truly novel. I suggest that they
bring this point out even more strongly in the abstract so that the reader will know what to look for in the paper. Some minor
suggestions follow, along with the pertinent locations in the manuscript.

Response: Thanks for your encouraging comments. Following your suggestion, we have highlighted better these findings in
the abstract. Below please find our point-to-point response to your other suggestions.

Figure 2b: What direction does “perpendicular” refer to? Horizontal and perpendicular to strike, or vertical?

Response: “perpendicular” refers to surface rupture displacements horizontally perpendicular to the fault strike, and we have
clarified that in caption of Figure 2b in the revised manuscript.

208-220: I'm a little confused by the rotation angles in this section. The fault changes its orientation by around 44 degrees as
one progresses along strike, and the stress changes its orientation by around 20 degrees over the same length. That means
that there must be a ~20 degree range in the angle of the stress with respect to the local fault orientation between different
portions of the fault. Do the authors still consider all these segments to be close to optimal orientation, even with such a
strong variation in the angle of the stress with respect to the fault?

Response: We thank your comment and agree our description is oversimplified. The southern-most section is almost entirely
well aligned to the regional stress field with an angle between the fault and σ_1 of ~23 degrees, while for the
northeastern most segment, the fault orientation is slightly less well aligned. The relative difference in the degree of fault
optimal alignment is quantified by the fault instability (higher values being more optimally aligned [Vavryčuk et al., 2013]).
This is illustrated by the stereonet plots in Fig. 7b-e by the location of the fault surface slip vectors shown as the white dots
and black vectors, where in the southern zone (Fig. 7b) they are in a region of the highest fault instability (yellow region, >
0.9), while the northern zone (Fig. 7c) the black vectors and white dots have migrated slightly away of it. We have now
clarified this in the text with the following:

“This illustrates that even though there is a ~44° change in the overall fault orientation along the entire length of the 320 km
long rupture, from ~east-west striking in the northeast region of the M7.8 rupture (segments A6 and A7) to ~SW striking in
the southwest (segments A1-A3), the fault surfaces are almost always optimally aligned because of the rotation of the
ambient pre-stress field along-strike (where the fault instability is highest along the southernmost section with the mean
value of fault patches of 0.98 and decreases slightly to 0.85 to the northeast).”

276-277: Could the authors clarify a little more how they explain the slip on the normal fault being in the direction opposite
that which would be expected from the local regional stress field? Are they saying that this small normal fault is largely
loaded by the nearby strike-slip fault, not by the regional tectonics?

Response: Yes, slip on the normal fault is not consistent with the regional stress field but is consistent with the stress changes

due to slip along the Surgu-Cardak fault. This poor alignment with the regional stress field can be seen in Fig. 7d and e) by
the small white dot in the eastern quadrant (~N90E), where the black vectors are in the opposite direction to those expected
from the regional stress tensor (gray vectors). This can be intuitively understood when looking at the red vectors in Fig. 7a
which depict the maximum horizontal stress direction, which is almost perpendicular to the normal fault orientation (where
thrust motion would be expected). This normal fault is therefore possibly associated with the tensional lobe caused by slip on
the Surgu-Cardak which would promote tension on that structure. We have now clarified this in the maintext with the
following:

*“Normal slip on this structure is however consistent with the stress changes due to slip along the S-C fault as this subsidiary*
*fault lies in an off-fault tensional stress lobe.”*

In conclusion, I think this is an excellent paper that will be of great interest to the community. I think it will be suitable for
publication with very minor revisions.

*Andrews, D. J., and Y. Ben-Zion (1997), Wrinkle-like slip pulse on a fault between different materials, Journal of*
*Geophysical Research, 102(1), 552-571.*

*Hu, F., D. D. Oglesby, and X. Chen (2020), The Near-Fault Ground Motion Characteristics of Sustained and Unsustained*
*Free Surface-Induced Supershear Rupture on Strike-Slip Faults, Journal of Geophysical Research: Solid Earth, 125(5),*
*doi:10.1029/2019JB019039.*

*Kyriakopoulos, C., B. Wu, and D. D. Oglesby (2021), Asymmetric topography causes normal stress perturbations at the*
*rupture front: The case of the Cajon pass, Geophysical Research Letters, 48(20), e2021GL095397.*

Response: Thanks again for your supportive words, and we have also included all the mentioned papers to our Reference list.

Reviewer #2 (Remarks to the Author):

The study obtained geodetic observations, and then performed slip model inversion and stress analysis for the 2023 Turkey
earthquake doublet. The authors concluded that the fault geometry matches with the stress tensor derived from focal
mechanisms, and argued that stress heterogeneity and free surface effect have an impact on the complex rupture process of
the earthquake. However, there are key issues in the geodetic data processing, finite fault inversions and stress analysis,
which have to be addressed before the manuscript could be considered for a publication.

Response: Thank you for your thorough review and valuable feedback on our manuscript regarding the geodetic
observations, slip model inversion, and stress analysis for the 2023 Turkey earthquake doublet. We understand the
significance of addressing these concerns to ensure the robustness and reliability of our findings, and we have revised (or
clarified) our manuscript accordingly.

The authors presented insufficient details for their geodetic data processing, i.e., on the unwrapped interferograms and offset
images. They also failed providing a self-consistent description about their 3D deformations and fault geometry/slip
inversion. Although they derived 3D deformations, they did not perform validation on the results, which show obvious
orbital errors. The authors derived 3D deformations but they didn't use them in fault geometry/slip inversion. They only used
SAR offset data to constrain the model. The model residuals also show obvious systematic error (the constant value). Near
some of the fault segments, the down-sampled data are very sparse, hence the resulted uncertainties are larger than the other
segments. The authors should perform more data uncertainty analysis as well as their impact on the model derived from
them.

Response: Regarding your concern about the insufficient level of details about geodetic data processing, please see our
response to your comments #1 and #2. We now validate the 3D deformations, using GNSS, see our response to your
comment #6. The 3D deformations were used to constrain surface rupture trace, i.e., to fix the varying strike angles. And due
to its relatively low accuracy in mapping deformation, we used only the line-of-sight (LOS) deformation of the ALOS-2
interferograms for the Bayesian estimation of subsurface fault dips. We have also included more supplementary figures to
better clarify the down-sampled data and enhance understanding of the model's uncertainties and robustness.

For stress inversion, the authors only provided the final stress directions but no uncertainties, which could be misleading and
difficult to interpret. It is not clear how can the authors derive robust stress fields using only 55 focal mechanisms without
uncertainty analysis.

Response: Following your comment, we have performed a comprehensive uncertainty analysis for the stress directions
derived from the 55 focal mechanisms, see our response to your comment #9.

In finite fault inversions, the authors did not rigorously separate the deformation of one event from another before using
them in the inversion. The waveform fitting quality is also not so good, in particular compared with those already published.

Response: We agree that it is not appropriate to perform inversion without separating deformation of one event from another.
Following previous studies [e.g., Liu et al., 2023], we have now excluded the InSAR measurements from finite source
inversion. The basic slip patterns remain almost unchanged. Besides, InSAR data is insensitive to rupture velocity, and our
results about rupture velocity do not alter. Besides, due to the removal of InSAR, we also see an improvement of waveform
fitting. For high-rate GNSS stations, the poor fits of vertical displacements are caused by the low signal-noise-ratio, and the
inaccurate velocity structure could also cause misfits.

Major Comments:

#1 The authors claimed they processed all ALOS-2 ScanSAR data, but they provided no details in supplemental materials.

The readers could be misled or confused if no wrapped or unwrapped interferograms are presented.

Response: We indeed processed ALOS-2 ScanSAR data from ascending track 184 and descending track 77 using alos2App.py
in ISCE. Both tracks fully cover the earthquake deformation area (see the interferograms shown regarding to your Comment
#4). The processing is fully automatic and described from lines 381-388 in the original manuscript. The only extra operation,
that is different from the standard processing of alos2App.py, is that we masked out the earthquake rupture region before phase
unwrapping. We have provided this detail in the revised manuscript:

*“We masked out the rupture areas before phase unwrapping to avoid phase unwrapping errors.”*

#2 Section “Remote Sensing” is also misleading. The title includes optimal images, but I cannot find any introduction about
Sentinel-2 data processing in the main text. There is only brief writeup in the supplement, but not connected with the main
text. This should be further improved.

Response: **Sorry for the absence of an introduction to Sentinel-2 data processing in the main text. They are methodological**
**details, and we feel it is appropriate to put that extra information in the supplements in order to shorten the article length. To**
**avoid any confusion, following your suggestion, we have now changed “Remote Sensing (InSAR and optical images)” to**
**“Processing of ALOS-2 measurements”.**

#3 The authors mentioned they used the range split-spectrum method to mitigate ionospheric noise. This is confusing
because range split spectrum’s accuracy is quite sensitive to unwrapping errors. If the fringes are very dense, SNAPHU may
fail, and range split-spectrum will lead to unavoidable errors in interferograms. The final fringes can be much denser or
dramatically sparser if the estimated ionospheric noise includes unwrapping errors. For these two earthquakes, the
unwrapping operation is very challenging. Please see the unwrapping operation provided by Xu et.al in their
Communications Earth and Environment preprint (<https://github.com/yunjunz/2023-Turkey-EQ>).

Response: We have carefully unwrapped the interferograms including the regular InSAR interferograms and subband
interferograms used for ionospheric corrections. We inspected all the unwrapped interferograms and are quite confident that
there are no significant phase unwrapping errors. The ruptured areas are not unwrapped. Note that it won't affect ionospheric
correction very much as long as the ruptured areas are small. For your interest, the details can be found at:

<https://github.com/isce-framework/isce2/discussions/676>

These procedures have been used to process many historical earthquake interferograms.

#4 The authors should give more details on phase unwrapping. There is no unwrapped interferograms in the manuscript and
supplement materials, but clear near-fault observations were shown in Figure S2 and S4. The results of the InSAR
observations in Fig.S2 and S4 should not be InSAR data, instead, they should be SAR range offset images.

Response: Phase unwrapping details can be found in the response to your comment #1. We confirm that we used ALOS-2
InSAR interferograms. Here we provide the original interferograms after ionosphere correction a), and the interferograms
after filtering and phase unwrapping b).
Ascending track 184, 220905-230220

Descending track d77, 220916-230217

It's clear that there are no significant phase unwrapping errors in the unwrapped interferograms. Likewise, there are also no
 significant phase unwrapping errors in subband interferograms, which is the reason why we could do a good ionospheric
 correction. We have now included the interferograms in the supplementary information.

#5 The north-south deformation in Fig.2a contains obvious stripe errors. That is due to offset images from Sentinel-2 are
 contaminated by orbital error. The authors only produced InSAR interferograms and some Sentinel-1 range offsets. They are
 all in the range direction which is not sensitive to the north-south direction. Therefore, only the north-south offset image
 from Sentinel-2 and the azimuthal offset from Sentinel-1 (these results should be very noisy although the authors didn't
 provide these azimuth offsets) contributes most to the final north-south deformation. Its orbital errors are all inherited to the
 final deformation. I strongly suggest the authors to perform Multiple Aperture Interferometry on ScanSAR data, which can
 improve the data quality in particularly for north-south deformation.

Response: Thanks for this insightful concern. However, we note that the long-wavelength offsets caused by orbital errors
 from Sentinel-1, do not significantly affect the estimates of surface fault slip (Fig. 2b) – which is what the 3D surface
 displacements (Fig. 2a) are used for - because fault slip is estimated from the discontinuity surface displacements across the
 surface rupture which occurs over a much shorter wavelength on the order of 100's of meters. In addition, the issue with the

north-south displacement direction is most pronounced in the northern region (around the M7.5 rupture) where we are
 lacking Sentinel-2 north-south displacement measurements due to poor correlation (see figure below, subpanel a)). However,
 as the M7.5 rupture is orientated in the east-west direction, the poorer constrained north-south direction does not
 significantly affect our estimates of fault slip along the M7.5 rupture. The north-south direction is well constrained along the
 M7.8 rupture as it is mostly constrained by the Sentinel-2 data where there is good correlation (see panel a) below).

#6 The authors should compare their 3D deformations to 3D GNSS offsets to validate the accuracy of 3D deformation.

Response: We note that the 3D surface displacement for the east-west and vertical components agree well with the GNSS
 from three sites that cover the surface displacement from our 3D displacement maps, but as expected the north-south
 component of surface displacement has a larger discrepancy due to the orbital errors from the Sentinel-1 pixel offsets and
 topographic artifacts from the Sentinel-2. However, this does not affect our estimates of fault slip (which is what the 3D
 surface displacement maps are used for in our study) because the M7.5 rupture is orientated almost exactly in the east-west
 direction, which has much better agreement with the GNSS. Moreover, as the fault slip is estimated from the differential
 motion across the rupture it is not affected by the longer wavelength errors that occur in the north-south direction.

Below shows the offsets from three representative GNSS stations where we have added the displacements from the M7.8

and 7.5 events and we have now included it in the supplementary information.

#7 The down-sampled deformation results (near Amanos fault) given in Fig.S2 and Fig.S4 are very sparse. The inverted dip
 angle should have large uncertainty here. The authors only give the MCMC uncertainty on their slip model, but they didn't
 perform any resolution or uncertainty tests for their slip model. These two tests are quite important to clarify if the data can
 resolve fault slip and/or geometry inversion robustly.

*Response: We apologize for the lack of clarity in our previous descriptions of the down-sampled data and inversion details.*

*To address this, we have provided a comprehensive supplement to elucidate these aspects.*

1) We have included additional supplementary figures to better illustrate the down-sampled data near the Amanos fault. The
 line-of-sight (LOS) deformation data from ALOS-2 interferograms have been down-sampled using the quadtree method for
 inversion. Full-resolution data maps (Figure S4) and data maps divided by the quadtree method (Figure S5) are available
 within the KITE software [Isken et al., 2017], which down-samples data based on the spatial gradient of deformation.
 Notably, the larger the deformation gradient, the denser the sampling. For down-sampling, 2416 quadtrees and 2687
 quadtrees were applied to ascending and descending tracks, respectively. It is evident that the down-sampled data effectively
 capture coseismic deformation. Excluding portions removed due to correlation loss in the original resolution maps, the
 down-sampled data points near the Amanos fault (segments A1 and A2, Figure 1) are not sparse compared to other areas
 near the fault. Additionally, the larger uncertainty of the dip angle near the Amanos fault may be attributed to the relatively
 small coseismic deformation in this area, making it less informative for determining the fault dip at depth.

2) We would like to clarify that our Bayesian estimation of the fault geometry was conducted assuming a uniform slip on
 each sub-fault (see Figure S6). This is the reason why the resolution test for the slip model was not applied directly. In this
 study, we employed the Bayesian approach with the MCMC sampling technique to estimate the posterior probability
 distributions of model parameters, governed by the data. The use of Bayesian approach aimed to characterize the
 uncertainties of parameters with their posterior probability distributions. In our Bayesian estimation of the fault geometry,

we modeled each fault segment as a rectangular fault plane with uniform slip (see Figure S6 below). The SAR 3D
deformation derived from pixel offsets can capture the fault surface rupture traces. Therefore, the fault strike and the length
along strike of each segment were set based on the SAR 3-D deformation. To invert for these parameters, we used the radar
line-of-sight (LOS) deformation obtained from ALOS-2 interferograms on both ascending and descending tracks, as well as
the GNSS offsets. We did not use SAR 3-D deformation because of its accuracy relatively poor compared with the ALOS-2
LOS deformation. Then the dip angle, the width along the dip, and the strike-slip and dip-slip components of all segments
were simultaneously sampled using the MCMC technique within the Bayesian framework. We summarized the standard
deviation of posterior samples for all parameters to characterize the parameter uncertainties (refer to Figure 3). Our preferred
model was chosen based on the median of posterior samples (Figure S6 below), which demonstrated a good fit to the data
(Figure S8 below).

We have accordingly revised section “Dip angles of each segment” in main text and section “Bayesian inversion of fault
geometry” in Methods, and included all figures in the supplementary material.

**Figure S4.** ALOS-2 LOS displacement (positive toward satellite) maps from the ascending track (a) and the descending
track (b), respectively. The green lines show the surface rupture traces inferred by the SAR 3D deformation.

Figure S5. Quadrees applied for down-sampling ALOS-2 LOS displacement maps for the ascending track (a) and the descending track (b), respectively. The longitude and latitude have been shifted with respect to the reference point (35°E, 36°N).

Figure S6. 3-D view of fault geometry and uniform slip on each segment, estimated by the medians of their posterior samples from the Bayesian inversion.

 **Figure S8.** The fit to the data for the Bayesian inversion of the fault geometry. (a) The horizontal GNSS observations (blue)
 versus the corresponding model predictions (red). The ALOS-2 line-of-sight observations (b and e), the corresponding model
 predictions (c and f), and the unmodeled residuals (d and g) from the ascending and descending tracks are color-coded in the
 same scale. Fault segments are marked with black solid lines. AZ, azimuthal direction. LOS, line of sight. Positive pixel
 values indicate motion towards the satellite.

**Reference:**

Isken, M., H. Sudhaus, S. Heimann, A. Steinberg, S. Daout, and H. Vasyura-Bathke (2017), Kite - Software for rapid
 earthquake source optimisation from InSAR surface displacement

 #8 Following #5, the aftershock catalog near A1 (Fig.1) shows shallow dip angle (at least smaller than 70 degree) but the
 inversion showed a result of ~80 degree. The authors should double check their data and the MCMC process. A resolution
 test would be needed as well.

Response: In this study, we utilized the relocated aftershock catalog for the initial 20 days from Ding et al. (2023). To
 enhance the visualization of the relationship between fault geometry and aftershock distribution, we now present a 3D view
 in Figure S7. This visualization illustrates that aftershocks near segment A1 are widely distributed in space. Our estimated
 dip angle for segment A1 is $80\pm 5^\circ$, and notably, this falls within the observed range of the aftershock distribution.
 We have included the figure in the supplementary material.

 Figure S7. 3-D view of the fault geometry color-coded with slip and the distribution of aftershocks (black dots, Ding et al.,
 2023). Segments A1, A2, A8 and A9 are labeled. The black arrow indicates the north direction. Red stars represent

hypocenters determined by the U.S. Geological Survey (USGS), GEOForschungsNetz (GEOFON), the Incorporated
Research Institutions for Seismology (IRIS) and the Turkey Disaster and Emergency Management Authority (AFAD).

Reference

Ding, H., Zhou, Y., Ge, Z., Taymaz, T., Ghosh, A., Xu, H., Irmak, T.S., Song, X., 2023. High-resolution seismicity imaging
and early aftershock migration of the 2023 Kahramanmaraş (SE Türkiye) MW7.9 & 7.8 earthquake doublet. *Earthquake
Science* 36, 417-432.

#9 The authors also present no detail on the stress inversion. More details should be given to verify if 55 pre-event (2007-
2020) focal mechanisms can derive robust principal stress directions. Please give sufficient details on how to derive the
stress tensor and the uncertainty, as well as on the diversity of slip vectors.

Response: Thank you for highlighting this concern. We have now provided the following text (Text S3 in supplementary)
that describes how we estimate the stress tensor, its uncertainty and determine the diversity of the slip vectors necessary for a
stable inversion.

The 3D deviatoric stress tensor is estimated by inverting the focal mechanisms from Guvercin et al. (2022) under the Wallace-
Bott assumption that slip is parallel to the shear stress (Michael, 1984). The stress inversion gives the orientation and shape of
the 3D deviatoric stress tensor but not its magnitude. The principal deviatoric stresses $\sigma_1, \sigma_2, \sigma_3$ are ordered from most to least
compressive. To resolve spatial variations of stress along the rupture we distinguish three zones along the M7.8 mainshock
rupture, making sure that each contains sufficient diversity of fault orientation to resolve the stress tensor according to the
criteria given by Hardebeck and Hauksson (2001). Where for all zones the angle of the slip vectors have an RMS > 30° from
the average (see histogram figure below which shows the distributions of each zone).

To invert the unit slip vectors for stress we use a L2 least squares inversion (Aster et al., 2011). To minimize overfitting of the
data and to constrain stress to be spatially smooth along the rupture we use a damping constraint to the inversion that penalizes
large changes of the stress orientation between neighboring zones (i.e., the gradients of the model vector between cells)
(Hardebeck & Michael, 2006). The uncertainties of the stress model are then estimated from bootstrapping via random
replacement of the original unit slip vectors.

Below shows the histogram of the RMS slip vector from the average orientation. Blue is the southern zone, orange is the
central zone and yellow is the northern zone.

Qualitatively, our stress tensors agree with those from Guvercin et al. (2022) and Gabriel et al. (2023) (for latter see figure
below) which both show a northeastward rotation of sigma-1 along the M7.8 rupture from the Amanos segment in the
southwest to the northeast. In addition, we note that the four stress zones we use along the M7.8 and M7.5 ruptures are larger
than those used in Guvercin et al. (2022), which is the study that we take the focal mechanism data from.

Figure below is from Gabriel et al. (2023) showing the stress orientation along the M7.8 and M7.5 ruptures.

(b)

Gabriel, A.A., Ulrich, T., Marchandon, M., Biemiller, J. and Rekoske, J., 2023. 3D Dynamic Rupture Modeling of the 6 February 2023, Kahramanmaraş, Turkey M w 7.8 and 7.7 Earthquake Doublet Using Early Observations. *The Seismic Record*, 3(4), pp.342-356.

#10 Figure S3 is not properly presented, with no figure caption. The strong motion station IDs could not be seen in the
figure. The waveform fitting quality in Fig.S3 and Fig.S5 are not so good.

Response: We apologize for the oversight in Figure S3, which was probably caused by converting word to PDF, and we have
now added figure caption and the strong motion station IDs. About the waveform fitting quality, see our response above.

#11 The authors simply truncated the geodetic deformation of one event from another (Fig. S2 and S4) in space, without
considering the actually impact of the deformation from one event to another, this is not appropriate.

Response: We have now excluded the InSAR data from finite source inversion and updated the slip models accordingly.
 Furthermore, following your suggestion, we have performed Jackknife test to assess the uncertainty of our finite source
 inversion models, see the Figures S11 and S14.

Reviewer #3 (Remarks to the Author):

Chen et al., utilized abundant geodetic and seismic data to investigate the rupture kinematics of the 2023 Kahramanmaras,
 Turkey M7.8 earthquake doublets. They propose that sub-faults involved in the M7.8 and M7.5 events align closely to
 optimal orientations of derived stress tensor from historical earthquake focal mechanisms. I think it's an interesting and
 consistent scientific insight to explain the initiation and propagation of large earthquake events, but I have several major and
 minor comments before the paper is accepted to publish.

Response: We are grateful for your positive comments on the scientific insight provided in our study. We have carefully
 considered and incorporated your comments into the revised manuscript. Below please find our point-by-point response.

Major comments:

1. Lines 219-220 and fig 7: Can you please show the strike angle of each principal stress axis and the mean slip vector? It's
 clearer to have a table that compares the fault strike angle and corresponding optimal orientation of each local stress tensor,
 assuming a static friction of 0.6 in your case.

Response: Thanks for your suggestion. Below are values of the mean fault strike from the slip model and the SHmax in
 degrees east of north, and we have supplemented it in Table S5.

	SOUTHERN ZONE	CENTRAL ZONE	NORTHERN ZONE
SHMAX (°)	355.7	2.8	20.6
MEAN FAULT STRIKE (°)	22.6	47.9	67.2

2. Fig 7 shows that most of fault traces of M7.8 events are optimally aligned with the pre-stress field, however, the NPF where
 the M7.8 earthquake initiated is not aligned with the "central" pre-stress field. Can you further explain why the mainshock
 initiated along a splay fault that doesn't have the optimal crack direction?

Response: It is not entirely clear if the NPF is misaligned to the regional 'central' pre-stress state or not as the fault strike of
 the NPF is not clear. Surface rupture observations (both field and pixel tracking) indicates it is ~N19°E, which would be highly
 mis-orientated to the regional 'central' stress state. However, relocated aftershocks from Ding et al. (2023) indicate a number
 of more NE orientated structures where the hypocenter is located, indicating a strike of N38°E (see figure below of the NPF-
 EAF region showing Ding et al. [2023] aftershocks with blue circles, red lines show the fault strike and SHmax [N2.8°E],
 green lines the mapped surface traces, and red-blue background colors are the north-south surface displacement). Under the
 current 'central' pre-stress state the angle between SHmax and a fault strike (presuming it is ~N38°E) would be 31.2, or a

static friction of 0.61. However, if the NPF does have an orientation of N19°E, then this would suggest a spatially heterogenous
stress field would be required for it to be optimally aligned.

**References**

*Ding, H., Zhou, Y., Ge, Z., Taymaz, T., Ghosh, A., Xu, H., Irmak, T.S. and Song, X., 2023. High-resolution seismicity*
*imaging and early aftershock migration of the 2023 Kahramanmaraş (SE Türkiye) MW7.9 & 7.8 earthquake*
*doublet. Earthquake Science, 36(6), pp.417-432.*

We have included the above statements in the main text and the figure in the supplementary information.

3. Your lines 246-247 state that the early transition to regional supershear rupture requires a high pre-stress on the fault. The
evidence from Weiss et al., 2020 suggests a 30-degree rotation of the first principal stress tensor (assuming strain rate and
stress have the same tensor). I am curious how such a stress axis rotation would allow for high initial shear stress
quantitatively?

Response: Thanks for pointing this out. The angle of SHmax with the C-S is ~60 degree at the epicentral location, which is
mis-orientated, and we agree that it is somewhat arbitrary to comment on whether the pre-stress amplitudes were high, given
that we only resolve the stress directions. However, considering we are merely suggesting that the observation of early
superstar indicates there was likely high pre-stress, without definitively asserting it, the current wording of the sentence still
appears acceptable.

4. Lines 251-252: In the case of 2019 M7.1 Ridgecrest earthquake, Jin and Fialko, 2020 reveals a scenario that the
mainshock rupture can nucleate on a small but more optimally oriented fault but propagated along less-favorably oriented
pre-existing faults subsequently due to dynamic weakening.

Response: We agree with the reviewer that dynamic weakening effects could explain rupture on segments B5 and B6 which
seem highly misaligned to the stress field we infer from focal mechanisms and the strain-rate field. We have now added two
sentences to the discussion section with the following:

*“Alternatively, nucleation along an optimally aligned fault segment (B3) that then induced strong dynamic weakening effects*
*could also explain the poorly aligned B5 and B6 segments, which was thought to occur for the 2019 Mw 7.1 Ridgecrest event*
*where rupture initiated on a small, optimally aligned fault segment that then propagated onto less-favorably orientated pre-*
*existing faults (Jin and Fialko, 2020). ”*

Minor comments:

1. Line 34: Please add the fault trace of 2020 Jan M6.7 Elazig earthquake into fig 1, because it’s also occurred along the East
Anatolian Fault (EAF).

Response: Done.

2. Line 174: Since the geodetic moment magnitude calculated from your slip model is 7.8, why did you assume the moment
magnitude of the second earthquake is 7.5 not 7.8?

Response: Apologies for any confusion. When we mention M7.5 for the second earthquake, it refers to the surface wave
magnitude estimated by USGS, rather than the moment magnitude from our inversion. Regarding the Turkey doublet, the
USGS surface wave magnitudes M7.8 and M7.5 are more widely used both in public and academic circles. In this context,
we are aligning with this convention.

3. Lines 149 & 176-177: The second event has a similar moment magnitude (M7.8) but a concentrated slip distribution
compared to the first one. Why does your slip model reveal the peak slip of both earthquakes is ~ 10 meters? According to
Jia et al., 2023 and Liu et al., 2023, both their slip models reveal that the peak slip is even larger than 12 m.

Response: The inversion of slip distribution is an underdetermined problem, typically requiring regularization to ensure
result stability. In this regard, smoothing factors must be set, and a stronger smoothing factor tends to result in a smaller peak
slip. Different research groups have varying preferences regarding how to choose smoothing factors, leading to variations in
peak slip values among different studies. Consequently, while some slip models suggest that the peak slip exceeds 12 m in
studies by Jia et al., 2023, and Liu et al., 2023, the other slip models from Tang et al., 2023, and Barbot et al., 2023, indicate
a peak slip of around 10 m.

4. Line 658: In figure 2a, why doesn't the fault trace of M7.5 event match with the polarity boundary of north-south
displacements?

Response: The polarity boundary of the north-south displacements that the reviewer refers to is an artifact due to removal of
north-south pixel offsets from the optical Sentinel-2 data. The optical pixel tracking in that region has very large topographic
noise, therefore we removed data in the region of the M7.5 rupture region where there is larger topographic relief. This
means the 3D surface displacement is mostly constrained by the radar offsets. Although the north-south component of
surface displacement is not as well constrained in this region compared to further south, our estimate of the fault slip (Fig.
2b) is not affected because the M 7.5 rupture is orientated in the east-west direction.

5. Figure 3: Why are dip angles of A8 and A9 fixed to 73 and 90 degrees, respectively? Are these two dip angles constrained
precisely by the relocated seismicity?

Response: Correct. We also referenced the hypocenters reported by USGS, GEOFON, IRIS and AFAD. To align with the
initiated fault segment, we fine-tuned the dip angle of segment A8 to ensure that the hypocenters from these four institutions
generally corresponded to the vicinity of the fault plane. In order to visually clarify the dip angle settings of segments A8 and
A9, we now present a 3D view depicting our fault geometry and aftershock distribution (refer to Figure S7).

We have added the figure in the supplementary material.

Figure S7. 3-D view of the fault geometry color-coded with slip and the distribution of aftershocks (black dots, Ding et al,

2023). Segments A1, A2, A8 and A9 are labeled. The black arrow indicates the north direction. Red stars represent

hypocenters determined by the U.S. Geological Survey (USGS), GEOForschungsNetz (GEOFON), the Incorporated

Research Institutions for Seismology (IRIS) and the Turkey Disaster and Emergency Management Authority (AFAD),

Reference

Ding, H., Zhou, Y., Ge, Z., Taymaz, T., Ghosh, A., Xu, H., Irmak, T.S., Song, X., 2023. High-resolution seismicity imaging
and early aftershock migration of the 2023 Kahramanmaraş (SE Türkiye) MW7.9 & 7.8 earthquake doublet. Earthquake
Science 36, 417-432.

6. Figure 7: Where are the gray vectors? The red symbols that represent the principal stress axes and white dots are too small
to see in those inset maps.

Response: Sorry for the unclear figure. We have re-plotted it to enhance visibility of the symbols. For your interest, we show
the pre-stress tensors as below:

REVIEWER COMMENTS

Reviewer #1 (Remarks to the Author):

I have read the response to my comments, as well as the revised manuscript. The authors have admirably implemented my suggestions, and as far as I am concerned, the manuscript is suitable for publication.

Reviewer #2 (Remarks to the Author):

The revised manuscript only partially address my comments. Based on the narrative of the manuscript, the key scientific contribution lies on the conclusion that the rupture of the earthquake doublet was controlled or “steered” by the prestress directions in the region. However, as also discussed in the last part of the paper, other factors, e.g. fault geometry, material contrast and frictional parameters etc., can also play a key role in governing the dynamic rupture of large earthquakes. Although I am not completely against using prestress conditions to explain the doublet, the evidences and the logic presented in the paper is not convincing, as elaborated below.

1. The authors claimed that they have inverted the pre-doublet principle stress directions using 55 background earthquakes. This is clearly a very important part of the story. However, I don't even see a map showing the distribution and focal mechanisms of these 55 events, there is also not a map to clearly show the zones that stress directions are inverted for.

2. In addition, it is not clear what is the uncertainty of these stress inversions, therefore I can't assess the robustness of the argument that the M7.8 event was favored by the stress condition and the M7.6 was not. The author presented a new supplement figure (Fig.S16) to show the “angle from average slip vector”. However, firstly, this is not an uncertainty analysis of stress inversion, and secondly, this is a confusing figure difficult to understand (e.g. what are the beach balls and their numbers within each zone? there are five colors shown in the figure but the legend only explain three).

3. Moreover, since the zone area is not shown, I can't assess what exactly are these principle stress (in Fig.7) are representing. For instance, is the stress direction near the nucleation fault branch representing the stress on the mainshock or the nucleation branch?

With above comments on the stress inversion, the focus on the paper is not so clear. The major portion of the technical work was conducted on deriving the finite fault model (still with quite some issues though, see more below), but the key scientific conclusion relied heavily on the stress inversion that has insufficient details. One can just collect the published rupture/slip models and get the same input for the stress analysis (Fig.7). To really verify the impact the prestress on the rupture, earthquake dynamic simulations will provide more quantitative and convincing supports, which was only briefly discussed at the end of the paper.

Technically, the revised manuscript still present quite a few issues in geodetic data processing, finite fault inversion and presentation details. While the authors made some attempts to engage

with my previous comments, the core problems weren't effectively resolved. I still have some major concerns :

Major concern 1:

In the first round of review, I gave some comments on the derived 3D surface deformations, especially for the stripe errors in the deformation maps. What a pity that they are still there. The stripe errors and noise are still clear in Fig.2a. I noticed some published 3D deformation results and share them with the authors. Space geodetic results can never be perfect but can be better after time goes. A year after the earthquake, the areas that were covered with snow in February-April 2023 have new, snow-free sentinel-2 images, which should be addressed in pursuit of better observations, but I haven't seen any updates.

Please see the below two deformations maps using the same data.

<https://zenodo.org/records/7966477>

<https://zenodo.org/records/8405014>

Major concern 2:

I am confused by the validation in Fig.S1. There are many GNSS stations around the East Anatolian Fault system. Nevada Geodetic Laboratory reported their coseismic offsets (<http://geodesy.unr.edu>). Why the authors only compare with three stations? Given the large discrepancy between the 3D deformations and GNSS offsets (Fig.S1), how to convince the readers of NC?

Major concern 3:

The authors tried to prove their interferograms with no unwrapping errors, and wrapped two interferograms into -20 rad to 20 rad, which is equivalent to -45 cm to 45 cm. This operation is quite confusing. It cannot provide any evidence of no unwrapping error. The fringes of those areas with large deformation exceeding a few of meters are very dense. How can we see them in this wrapped phase map? Why don't you show the original interferogram, filtered interferogram, unwrapped interferogram respectively? The robustness of the results could be verified by presenting them in a variety of formats, i.e., a direct comparison with GNSS offsets project into Line-of-Sight direction.

Major concern 4:

The authors present no details of fault geometry for the Mw7.5 event. I can just capture the geometry of Mw7.8 event from Fig.S6-7, but nothing for Mw7.5 event.

I can see the dipping angle of segment B5 from Fig.3, but it is in a dramatically shallow dipping angle of 54 degrees. How can you explain the discrepancy between the concentrated aftershocks and the nearly vertical fault geometry in the published references? I list all of them below. Please double check.

Ren et.al ,Supershear triggering and cascading fault ruptures of the 2023 Kahramanmaraş, Türkiye, earthquake doublet, Science, 383, 6680, (305-311), (2024).

Jia et al. ,The complex dynamics of the 2023 Kahramanmaraş, Turkey, Mw 7.8-7.7 earthquake doublet.Science381,985-990(2023).DOI:10.1126/science.adi0685

Major Concern 5:

The authors presented an RMS of slip vectors from their used average orientation in Fig.S16. The

deviation of slip vectors can reach more than 80 degrees. Do the authors think the derived stress orientations are reliable? You should give an additional uncertainty result from bootstrapping test like the method using in the reference below.

Milliner, C. W. D., Aati, S., & Avouac, J. P. (2022). Fault friction derived from fault bend influence on coseismic slip during the 2019 Ridgecrest Mw 7.1 mainshock. *Journal of Geophysical Research: Solid Earth*, 127, e2022JB024519

Such large RMS with few samples in Fig.S16 is not convincing. These stress fields are the foundation of your story. Please give more validation.

Major Concern 6:

The fault trace in green lines (Fig.1, Fig.S4) and in black lines (Fig.7) are the same as the mapped simple fault trace from Reitman. et.al. Please see

<https://www.sciencebase.gov/catalog/item/644ad9afd34e45f6ddccf736>

The authors need to declare how they derived this fault trace from 3D deformations that even have no near-fault observations (Fig.S4).

Major concern 7:

The authors used multiple triangle source time function in the finite fault inversion but did not show any details on the sub-fault source time functions. I am not sure how the rupture speed is defined in the multiple triangle source time function manner, is it the beginning or corresponding to the triangle with the peak amplitude?

Although GPS and strong motion waveform fits are presented, I am not sure how did the author align the synthetics and data? Do they align from origin time or first P-wave arrival? It is clear that the arrival time of the S-wave pulses place the most important constraints on the rupture speed (e.g. Fig.S13). Also for Fig.S13, I can't find all the stations in the map for high-rate GPS, and can't see all the waveform fits for the strong motion stations. For Fig.S15, what is the rupture speed for the eastward rupture?

Reviewer #3 (Remarks to the Author):

I appreciated that the authors addressed all my questions clearly, and I think the paper presents an interesting and consistent scientific insight to explain the initiation and propagation of large earthquake events when I firstly read the paper. I think the paper can be published as it is now.

Reviewer #4 (Remarks to the Author):

The manuscript of Chen et al. reexamines the rupture dynamics of the 2023 Kahramanmaras, Turkey earthquake doublets combining strong motion, GNSS InSAR, optical imagery datasets. Findings of the rupture kinematics, e.g., coseismic slip distribution and rupture propagation, are

more or less similar to those earlier results. I agree with the previous reviewers that the discussion of rupture kinematics in the context of the pre-earthquake stress field makes this study more interesting and important. I have to admit that I read through the manuscript very quickly, but I did have a pretty careful look at the author's response to previous comments and suggestions. I think that the authors have done a sufficiently good job answering the questions and revised the manuscript accordingly. I therefore recommend this manuscript to be accepted as is (maybe after a few very minor edits as suggested below)

Line: 201-202: 'and the two' --- incomplete sentence?

Color range for aftershocks shown in Figure 1 appears to be quite narrow, making it hard to make out the main features of the aftershock depth distribution. Changing it to another color template It may also be informative to plot out the aftershock distribution alongside the inverted slip distribution (Figure 4 and 5), although I know the discussion of the relationship between mainshock and aftershocks is not the focus of this study.

It is up to the authors, but I feel that it may be better to combine Figure 4 and 5 into the same figure.

**REVIEWER COMMENTS**

Reviewer #1 (Remarks to the Author):

I have read the response to my comments, as well as the revised manuscript. The authors have
admirably implemented my suggestions, and as far as I am concerned, the manuscript is
suitable for publication.

**Response:** We sincerely appreciate the reviewer's time and effort in assessing our manuscript
and providing insightful comments.

Reviewer #2 (Remarks to the Author):

The revised manuscript only partially address my comments. Based on the narrative of the
manuscript, the key scientific contribution lies on the conclusion that the rupture of the
earthquake doublet was controlled or “steered” by the prestress directions in the region.
However, as also discussed in the last part of the paper, other factors, e.g. fault geometry,
material contrast and frictional parameters etc., can also play a key role in governing the
dynamic rupture of large earthquakes. Although I am not completely against using prestress
conditions to explain the doublet, the evidences and the logic presented in the paper is not
convincing, as elaborated below.

**Response:** We thank the reviewer for the continued engagement with our manuscript and for
providing further feedback. We have made further revisions to emphasize the focus of our
paper. We believe that our manuscript now clearly outlines the key influence of principal stress
rotation in directing fast, supershear rupture along complex fault geometries. We have provided
observational evidence supporting that claim and we refer to dynamic simulations presented
by other groups to support this conclusion.

While we appreciate your suggestion regarding the need for additional earthquake dynamic
simulations, we would like to emphasize that our interpretation is based on numerical
simulations conducted in previous studies (e.g., Oglesby et al., 1998, 2000; Kyriakopoulos et
al., 2021), which provide theoretical insights into the roles of stress orientation and dip angle
of faults in causing supershear rupture and large slip. Furthermore, specific dynamic rupture
models have been performed for the events discussed in our paper, contributing to our
understanding of the observed kinematics (e.g., Gabriel et al., 2023; Jia et al., 2023). These
simulations do assume heterogeneities of stress consistent with the rotation we described in our

study, but the authors didn't point to the importance of this feature for their simulations to
reproduce observations.

1. The authors claimed that they have inverted the pre-doublet principle stress directions using
55 background earthquakes. This is clearly a very important part of the story. However, I don't
even see a map showing the distribution and focal mechanisms of these 55 events, there is also
not a map to clearly show the zones that stress directions are inverted for.

**Response:** As stated in the supplements (Text S3), the inversion for stresses used the focal
mechanism catalogue of Guvercin et al. (2022). In figure S20, we provide statistics to show
these stress zones provide sufficient diversity in the mechanism to determine the stress tensor,
as according to the synthetic tests conducted by Hardebeck and Hauksson (2001), which
requires a minimum of 30deg. in the RMS. However, to illustrate the location of the three zones
along the rupture we have now included a figure in the supplements (Figure S19).

Figure S19. Map illustrating the location and extent of the three stress zones (green lines),
 with the coseismic rupture traces mapped (blue lines) and focal mechanisms from Guvercin
 et al. (2022).

2. In addition, it is not clear what is the uncertainty of these stress inversions, therefore I can't
 assess the robustness of the argument that the M7.8 event was favored by the stress condition
 and the M7.6 was not. The author presented a new supplement figure (Fig.S16) to show the
 "angle from average slip vector". However, firstly, this is not an uncertainty analysis of stress
 inversion, and secondly, this is a confusing figure difficult to understand (e.g. what are the
 beach balls and their numbers within each zone? there are five colors shown in the figure but
 the legend only explain three).

**Response:** In our response to concern #5 we now provide two supplementary figures which
shows the uncertainty of our stress inversion and goodness of fit, which allows an assessment
of the robustness of the stress tensor estimates. We have also improved Fig. S16, which
contained a confusing color scheme, where three distributions were plotted on top of one-
another that made it appear that there were five colors. Below is our updated figure which
separates these distributions and makes it clearer. As the reviewer mentioned this is not an
uncertainty analysis but we provide it to demonstrate that we have sufficient diversity in our
focal mechanism catalogue to robustly estimate the stress tensor (which we now state clearly
in the supplements and figure caption – we elaborate on this point in our response to concern
#5 below). In our response to point #5 we detail how we estimate the uncertainty and that our
stress tensor estimates are robust.

Below is our new supplementary figure that shows the misfit estimates (the angular misfit) that
can give a sense of the goodness of fit of our stress tensors to the focal mechanisms in each of
the three zones.

Fig. S22. Distributions showing the angular misfit between the predicted and observed focal
 mechanism from our stress tensor inversion for the three zones.

3. Moreover, since the zone area is not shown, I can't assess what exactly are these principle
 stress (in Fig.7) are representing. For instance, is the stress direction near the nucleation fault
 branch representing the stress on the mainshock or the nucleation branch?

With above comments on the stress inversion, the focus on the paper is not so clear. The major
 portion of the technical work was conducted on deriving the finite fault model (still with quite
 some issues though, see more below), but the key scientific conclusion relied heavily on the
 stress inversion that has insufficient details. One can just collect the published rupture/slip
 models and get the same input for the stress analysis (Fig.7). To really verify the impact the
 prestress on the rupture, earthquake dynamic simulations will provide more quantitative and
 convincing supports, which was only briefly discussed at the end of the paper.
 Technically, the revised manuscript still present quite a few issues in geodetic data processing,
 finite fault inversion and presentation details.

**Response:** Following your suggestion, we have delineated the zone area in the updated Figures
 7 and S19. Regarding the focus of our paper and dynamic rupture simulation, please refer to
 our response to your previous comment above. For your concerns about geodetic data
 processing, finite fault inversion, and presentation details, we have addressed these issues in
 our point-by-point response below.

While the authors made some attempts to engage with my previous comments, the core
 problems weren't effectively resolved. I still have some major concerns :

Major concern 1:

In the first round of review, I gave some comments on the derived 3D surface deformations, especially for the stripe errors in the deformation maps. What a pity that they are still there. The stripe errors and noise are still clear in Fig.2a. I noticed some published 3D deformation results and share them with the authors. Space geodetic results can never be perfect but can be better after time goes. A year after the earthquake, the areas that were covered with snow in February-April 2023 have new, snow-free sentinel-2 images, which should be addressed in pursuit of better observations, but I haven't seen any updates.

Please see the below two deformations maps using the same data.

<https://zenodo.org/records/7966477>

<https://zenodo.org/records/8405014>

Response: We have now improved our geodetic imaging data by including optical image pairs that were not acquired during periods of snow cover. However, we note that this does not significantly alter the fault slip measured along the surface ruptures from what we already presented in our previous submission (Fig. 2b and c). The reason is that the original north-south measurements of surface motion, which were the most poorly constrained, were noisiest in the northern region where the Mw 7.6 rupture occurred (due to significant snow and relief in the mountainous terrain that can induce large topographic artifacts). However, because the Mw 7.6 rupture is orientated almost exactly east-west, measurements of fault slip are largely insensitive to the noisy north-south component of motion and as expected the updated results we present now have not significantly changed compared to what we showed in our prior submission. In addition, the use of stacked profiles which averages the surface motion over a finite along-fault swath width also helps suppress any noise in the surface displacement maps. Therefore, it is not surprising that the fault slip measurements have not changed substantially. Nonetheless, we now include our updated 3D surface displacement and fault slip results in our revised submission.

To demonstrate that our new fault slip measurements from the updated 3D deformation results
are not contaminated by any noise in our data we have compared them to two other published
studies (Karabulut et al., 2023; Ma et al., 2024) and a dataset provided by the UK research
group COMET. We find our results agree very well with all three of them, with correlation
coefficients ranging from 0.88-0.95. This has been supplemented in Figure S2.

We note that the surface fault slip measurements are only shown in the analysis to illustrate the
 general variation of slip along the rupture, and they are not used in the stress comparison or as
 a constraint for the finite-fault slip inversion.

Reference:

Karabulut H, Güvercin SE, Hollingsworth J, Konca AÖ. Long silence on the East Anatolian
 Fault Zone (Southern Turkey) ends with devastating double earthquakes (6 February 2023)
 over a seismic gap: implications for the seismic potential in the Eastern Mediterranean region.
 Journal of the Geological Society. 2023 May 5;180(3):jgs2023-021.

146 Ma Z, Li C, Jiang Y, Chen Y, Yin X, Aoki Y, Yun SH, Wei S. Space geodetic insights to the
147 dramatic stress rotation induced by the February 2023 Turkey-Syria earthquake doublet.
Geophysical Research Letters. 2024 Mar 28;51(6):e2023GL107788.

Major concern 2:

I am confused by the validation in Fig.S1. There are many GNSS stations around the East
Anatolian Fault system. Nevada Geodetic Laboratory reported their coseismic offsets
(<http://geodesy.unr.edu>). Why the authors only compare with three stations? Given the large
discrepancy between the 3D deformations and GNSS offsets (Fig.S1), how to convince the
readers of NC?

**Response:** We originally compared with only three GNSS stations because these were the only
solutions that were initially made available by UNR, which were derived using rapid orbits.
However, now that final solutions and additional stations have been made available, we have
now included them.

http://geodesy.unr.edu/news_items/20230213/us6000jllz_final5min.txt

Below shows our updated Figure S1 with comparison of the newly derived 3D deformation
result against eight GNSS stations with significant co-seismic displacements, the consistency
between GNSS is reasonably well.

**Figure S1.** (Left) Co-seismic east-west deformation inverted from the Sentinel-1 and Sentinel-
2 satellite image offsets. (Right) Comparisons between the coseismic surface displacement

from the 3D deformation geodetic imaging data (shown in Fig. 2a) and the GNSS records from
University of Nevada Reno (Blewitt et al., 2018).

Reference

Blewitt, G., W. C. Hammond, and C. Kreemer (2018), Harnessing the GPS data explosion for
interdisciplinary science, *Eos*, 99, <https://doi.org/10.1029/2018EO104623>.

Major concern 3:

The authors tried to prove their interferograms with no unwrapping errors, and wrapped two
interferograms into -20 rad to 20 rad, which is equivalent to -45 cm to 45 cm. This operation is
quite confusing. It cannot provide any evidence of no unwrapping error. The fringes of those
areas with large deformation exceeding a few of meters are very dense. How can we see them
in this wrapped phase map? Why don't you show the original interferogram, filtered
interferogram, unwrapped interferogram respectively? The robustness of the results could be
verified by presenting them in a variety of formats, i.e., a direct comparison with GNSS offsets
project into Line-of-Sight direction.

**Response:** Since phase unwrapping errors are usually sharp discontinuities, rewrapping the
unwrapped phase with a larger period is an effective way to inspect the phase unwrapping
errors. If there is a chunk of image that has phase unwrapping errors, there will be a sharp
discontinuity between this chunk of image and the neighboring chunks. This method has been
used by us and many others to check numerous unwrapped interferograms. Per your request,
here we present the original and filtered interferograms, as well as the unwrapped
interferograms without any phase wrapping. Figures S3 and S4 have been updated accordingly.

Ascending track 184, 220905-230220

-3.14 (rad) 3.14 -3.14 (rad) 3.14 -200 (rad) 200

Descending track 77 220916-230217

-3.14 (rad) 3.14 -3.14 (rad) 3.14 -200 (rad) 200

We compared InSAR results with GNSS data near the ruptures. These comparisons do not
 reveal phase unwrapping errors. Below are the results. In the figures, the circles represent GPS
 stations, and the colors inside them denote GPS displacements projected into radar line of sight
 directions. The arrows denote horizontal GPS displacements. The numbers beside the GPS
 station names denote the differences between InSAR and GPS measurements. All
 measurements are in meters. Clearly InSAR is in good agreement with GPS.

Displacement (m) Displacement (m)
 Ascending track 184, 220905-230220 Descending track 77 220916-230217

 Major concern 4:

The authors present no details of fault geometry for the Mw7.5 event. I can just capture the
 geometry of Mw7.8 event from Fig.S6-7, but nothing for Mw7.5 event.

I can see the dipping angle of segment B5 from Fig.3, but it is in a dramatically shallow dipping
 angle of 54 degrees. How can you explain the discrepancy between the concentrated
 aftershocks and the nearly vertical fault geometry in the published references? I list all of them
 below. Please double check.

 Ren et.al ,Supershear triggering and cascading fault ruptures of the 2023 Kahramanmaraş,
 Türkiye, earthquake doublet, Science, 383, 6680, (305-311), (2024).

Jia et al. ,The complex dynamics of the 2023 Kahramanmaraş, Turkey, Mw 7.8-7.7 earthquake
doublet.Science381,985-990(2023).DOI:10.1126/science.adi0685

**Response:** Regarding the display of fault geometry for the M7.5 event, we have revised Figs.
S7 and S8 , and enhanced the 3D visualization to better illustrate the fault geometry for the
M7.5 event. The updated Figs. S7 and S8 are provided in the supplementary information.

In terms of the dipping angle estimate of segment B5 in the M7.5 event, we wish to highlight
the novelty behind our estimation of the fault geometry using Bayesian inversion of geodetic
data. For the polished paper as you listed, Jia et al., (2023) set the sub-surface dipping angles
of all segments in the M7.5 event to be 70° based on the distribution of aftershocks. Ren et.al
(2024) constructed the sub-surface fault geometry by combining the multi-point source (MPS)
inversion with the locations of aftershocks. However, their MPS inversion did not include
segments east to the epicenter (segments B4 and B5). They treated the dipping angle of
segments B4 and B5 in the M7.5 event to be sub-vertical. We also noted other published papers
that inferred the fault geometry with aftershock distributions. For example, Liu et al. (2023)
inferred the sub-surface dipping angles of fault segments based on the locations of aftershocks,
estimating the dipping angle of their easternmost segment (B5) to be ~60°. The sub-surface
dipping angle estimates of fault segments by aftershock distributions are controversial. The
distribution of aftershocks is quite complex during the 2023 Kahramanmaraş earthquakes, and
the estimation of the sub-surface dipping angles depends on which spatial and temporal ranges,
magnitudes and depths of aftershocks are chosen to evaluate the fault geometry. He et al. (2023)
treated the fault of the M7.5 event as one segment and estimated its dipping angle by a grid
search of the data fitting of geodetic inversion. Their result shows a 59° dipping angle for the
M7.5 event. To address the possible fault dipping irregularities along strike during the M7.5
event, in this study, with Bayesian inversion of geodetic deformation data, we explored variable
fault dipping angles at different fault segments, which are crucial to estimating rupture
propagation. We divided the fault of M7.5 event into six segments along strike, and the
simultaneous estimation of six fault segments from geodetic deformation data points out that
segment B5 has the angle estimate of $54 \pm 4^\circ$. We therefore provided the independent estimate
of fault dipping irregularities along strike from geodetic inversion, rather than the inference of
aftershock locations. We present the comparison of the estimated geometry of segment B5 with
aftershock locations (See the figure below). It is indicated that the estimated fault geometry
remains within the spatial range of aftershock locations.

 3-D view of fault geometry of segment B5, color-coded with slip and the distribution of
 aftershocks (black dots, Ding et al, 2023). The black arrow indicates the north direction.

 We have clarified that in the revised manuscript.

 Reference

Ding, H., Zhou, Y., Ge, Z., Taymaz, T., Ghosh, A., Xu, H., et al. (2023). High-resolution
 seismicity imaging and early aftershock migration of the 2023 Kahramanmaraş (SE Türkiye)
 MW7.9 & 7.8 earthquake doublet. *Earthquake Science*, 36(6), 417-432.

 He, L., Feng, G., Xu, W., Wang, Y., Xiong, Z., Gao, H., & Liu, X. (2023). Coseismic
 Kinematics of the 2023 Kahramanmaraş, Turkey Earthquake Sequence From InSAR and
 Optical Data. *Geophysical Research Letters*, 50(17).

 Liu, C., Lay, T., Wang, R., Taymaz, T., Xie, Z., Xiong, X., et al. (2023). Complex multi-fault
 rupture and triggering during the 2023 earthquake doublet in southeastern Türkiye. *Nat*
 *Commun*, 14(1), 5564.

 Major Concern 5:

The authors presented an RMS of slip vectors from their used average orientation in Fig.S16.
 The deviation of slip vectors can reach more than 80 degrees. Do the authors think the derived
 stress orientations are reliable? You should give an additional uncertainty result from
 bootstrapping test like the method using in the reference below.

Milliner, C. W. D., Aati, S., & Avouac, J. P. (2022). Fault friction derived from fault bend
influence on coseismic slip during the 2019 Ridgecrest Mw 7.1 mainshock. *Journal of*
*Geophysical Research: Solid Earth*, 127, e2022JB024519

Such large RMS with few samples in Fig.S16 is not convincing. These stress fields are the
foundation of your story. Please give more validation.

**Response:** We have shown that the derived stress orientations are reliable. As explained in
Text S3 and illustrated by Fig. S20, we demonstrate that we can reliably estimate the stress
tensors. We have made this determination by estimating the RMS difference of the mechanisms,
finding that they exceed the 30 degree minimum value that was stated by Hardebeck and
Hauksson (2001) from synthetic tests to be a requirement to accurately estimate the stress
tensor. This minimum value is necessary to show the focal mechanism catalogue has sufficient
diversity in its orientation to reliably estimate the stress tensors. Fig. S20 shows the distribution
and values of the RMS difference which exceed 30 degrees for all 3 of the stress zones. This
point is stated in Text S3 by the following:

*"To resolve spatial variations of stress along the rupture we distinguish three zones along the*
*M7.8 rupture, making sure that each contains sufficient diversity of fault orientation to resolve*
*the stress tensor according to the criteria given by Hardebeck and Hauksson [2001]. Where*
*for all zones the angle of the slip vectors have an RMS > 30° from the average (see Figure S20*
*which shows the distributions of each zone)."*

We further clarify this point we have added to the supplements in Fig. S20 the following:

*"This shows that each stress zone meets and exceeds the minimum requirement of having a 30°*
*RMS difference from the mean mechanism orientation (Hardebeck and Hauksson, 2001). This*
*demonstrates that each zone has a sufficient diversity in mechanism orientation that is required*
*to reliably estimate the stress tensors."*

Second, our stress orientations agree qualitatively well with that of Guvercin et al. (2022)
which also shows an Andersonian strike-slip stress state along the rupture and a north-south
orientation of σ_1 in the southern region of the M7.8, which rotates clockwise northwest
along the M7.8 rupture, with σ_1 also having a NNE orientation at the northeastern
termination of the M7.8.

For the estimate of the stress tensor and its uncertainty our results do use the bootstrapping
analysis as suggested by the reviewer, which is the same as that of Milliner et al. (2022). We
now show the stress tensors and their uncertainties estimated from the bootstrapping in Fig.
S21.

Fig. S21. Illustration of the principal stresses and their uncertainties for the four zones plotted
in lower hemisphere stereonets.

Also see our response to the initial points raised by the reviewer where we have provided two
supplementary figures that show each zone contains a sufficient diversity in focal mechanism
orientation to robustly estimate the stress tensor and misfit statistics.

Major Concern 6:

The fault trace in green lines (Fig.1, Fig.S4) and in black lines (Fig.7) are the same as the
mapped simple fault trace from Reitman. et.al.

Please see <https://www.sciencebase.gov/catalog/item/644ad9afd34e45f6ddccf736>

The authors need to declare how they derived this fault trace from 3D deformations that even
have no near-fault observations (Fig.S4).

**Response:** We thank the reviewer for making us aware that there is a missing reference. The
fault traces in Figs. 1 and 7 are indeed from Reitman et al. (2023), and we have now added the
reference appropriately in each of the figure captions where it is shown.

However, we note that this analysis does include near-fault observations (Fig. 2b), which are
measured from the 3D result shown in Fig. 2a and is constrained by pixel offsets from the radar
and optical data. But this has insufficient spatial resolution to capture the fine-scale details of
the rupture trace that was mapped to higher detail by the Reitman et al. (2023) study using
high-resolution optical satellite imagery.

Reitman, N.G., Briggs, R.W., Barnhart, W.D., Thompson Jobe, J.A., DuRoss, C.B., Hatem,
337 A.E., Gold, R.D., Akçiz, S., Koehler, R.D., Mejstrik, J.D., Collett, C., 2023, Fault rupture
mapping of the 6 February 2023 Kahramanmaraş, Türkiye, earthquake sequence from satellite
data (ver. 1.1, February 2024): U.S. Geological Survey data release,
<https://doi.org/10.5066/P985I7U2>.

Major concern 7:

The authors used multiple triangle source time function in the finite fault inversion but did not
show any details on the sub-fault source time functions. I am not sure how the rupture speed is
defined in the multiple triangle source time function manner, is it the beginning or
corresponding to the triangle with the peak amplitude? Although GPS and strong motion
waveform fits are presented, I am not sure how did the author align the synthetics and data?
Do they align from origin time or first P-wave arrival? It is clear that the arrival time of the S-
wave pulses place the most important constraints on the rupture speed (e.g. Fig.S13). Also for
Fig.S13, I can't find all the stations in the map for high-rate GPS, and can't see all the waveform
fits for the strong motion stations. For Fig.S15, what is the rupture speed for the eastward
rupture?

**Response:** Per your request, we show the sub-fault source time functions in Figures S12 and
S16, see below for a reference.

**Figure S12.** Subfault source time functions for the M7.8 event, and red star denotes

epicenter. a) represents the splay fault NPF and b) shows the main EAF strand.

**Figure S16.** Subfault source time functions for the M7.5 event, and red star denotes

epicenter. a) represents the N-S striking B6 fault and b) shows B1 to B5 main strand.

Given that we use standard methods to determine the finite source model we provide only the

minimum information needed in the Method section and refer readers to the literature for details.

We adopted multi-time-window method (Hartzell and Heaton, 1983), which determines the

spatial and temporal slip distribution on a fault plane via a linear expression using successive

time windows (typically triangle source functions with half-duration overlap) propagating at a

predefined constant velocity. Specifically, we synthesized the GPS and strong motion full

waveforms using the F-K methodology (Zhu and Rivera, 2002) based on the local velocity

structure and aligned the waveforms to the earthquake origin time, with 50 epochs in advance.
For each subsequent window, the beginning times are lagged sequentially by the rise time. To
find the optimal rupture speed, we employed a trial-and-error method based on the waveform
fits. Ideally, P-waves and S-waves should be weighted equally. However, in near-field
observations, it is quite challenging to clearly separate P and S waves. Furthermore, high-rate
GNSS data is notably noisy, especially the vertical components, complicating the detection of
P-wave arrivals. Additionally, a very precise velocity structure is needed to model the
synthesized P waves reliably, which is not always available. As a result, S-wave pulses are the
dominant contribution to determining rupture velocity, which is also the case for other non-
linear inversion methods that treat rupture velocity as a variable.

We apologize for the previous inconsistencies between station distribution and waveforms.
This issue has now been addressed, and the station distribution plot in Figure 5 has been
updated. Regarding Figure S15, the rupture speed for the eastward rupture is 2.8 km/s, which
we have now clarified in the figure caption.

Reviewer #3 (Remarks to the Author):

I appreciated that the authors addressed all my questions clearly, and I think the paper presents
an interesting and consistent scientific insight to explain the initiation and propagation of large
earthquake events when I firstly read the paper. I think the paper can be published as it is now.

**Response:** Thank you again for taking the time to review our paper. Your positive feedback is
greatly appreciated.

Reviewer #4 (Remarks to the Author):

The manuscript of Chen et al. reexamines the rupture dynamics of the 2023 Kahramanmaras,
Turkey earthquake doublets combining strong motion, GNSS InSAR, optical imagery datasets.
Findings of the rupture kinematics, e.g., coseismic slip distribution and rupture propagation,
are more or less similar to those earlier results. I agree with the previous reviewers that the
discussion of rupture kinematics in the context of the pre-earthquake stress field makes this
study more interesting and important. I have to admit that I read through the manuscript very
quickly, but I did have a pretty careful look at the author's response to previous comments and
suggestions. I think that the authors have done a sufficiently good job answering the questions

and revised the manuscript accordingly. I therefore recommend this manuscript to be accepted
as is (maybe after a few very minor edits as suggested below)

**Response:** Thank you for your evaluation of our manuscript. We appreciate your
acknowledgment of our efforts in addressing previous comments and suggestions. We'll
ensure that any suggested revisions are carefully implemented to enhance the clarity and
coherence of our findings.

Line: 201-202: 'and the two' --- incomplete sentence?

**Response:** Thank you for pointing it out. We have revised it.

Color range for aftershocks shown in Figure 1 appears to be quite narrow, making it hard to
make out the main features of the aftershock depth distribution. Changing it to another color
template It may also be informative to plot out the aftershock distribution alongside the inverted
slip distribution (Figure 4 and 5), although I know the discussion of the relationship between
mainshock and aftershocks is not the focus of this study.

**Response:** We have replaced it by a wider color range. See the updated Figure 1. We have also
plotted the aftershock distribution alongside the inverted slip distribution in Figs. 4 and 5.

It is up to the authors, but I feel that it may be better to combine Figure 4 and 5 into the same
figure.

**Response:** Thank you for your constructive suggestions. When our manuscript was at its
initial stage, we did combine Figures 4 and 5 into one plot, but we found that the combination
made the plot looked very crowded. As a result, we prefer showing them separately.

REVIEWERS' COMMENTS

Reviewer #2 (Remarks to the Author):

The revised manuscript addresses part of my previous comments. But some of the critical points were still dodged by the authors.

1. Major concern on pre-stress heterogeneity from foreshocks

The number of foreshocks (55) that were used to derive the prestress orientation is too limited (Fig.S19), in particular when considering the size of the area ($\sim 300\text{km} \times 300\text{ km}$) and the complexity of the fault geometry. The way that the earthquakes are grouped is also questionable, for instance, in the southern segment, many of the events are quite far away from the rupture (or plate boundary), it is not clear if these earthquakes can represent the stress condition of the M7.8 rupture; in the M7.6 event area (the yellow polygon), most of the events are actually located along the northern fault segment of the M7.8 earthquake, how can these events represent the stress field for both the M7.6 event and the northern segment of M7.8 event? The main conclusion of the paper is based on the stress orientation derived from these 55 earthquakes, it is critical to show that the stress field they obtained robustly represent the stress field that governed the rupture of the doublet. I notice there are other recent publications on the stress field, it would be helpful that the authors compare their results with those derived from other methods.

Figure S19.

I also have questions on the stress inversion results (Fig.S21). Take the Surgu-Cardak region (the yellow polygon in Fig.S19) as an example, I use the same 14 focal mechanisms from [Güvercin et al., 2022], the same source as the authors obtained the focal mechanism, and perform similar stress inversion with uncertainty estimated by a bootstrapping method (similar as the authors did). If I set the friction to 0.6, again the same as the authors did, the resulted stress orientation is shown in upper panels of Fig.1, where the uncertainty of sigma-3 is much larger than that of sigma-1 and sigma-2. Even for sigma-1 and sigma-2, the uncertainty is much larger than that shown in Fig.S21d, no mention that if one allows a variable friction coefficient on the fault (lower panels in Fig.1), the uncertainty of stress inversion is getting even larger. With such strong contrasts and small number of foreshocks, I don't understand how the authors can obtain such tight uncertainties in their inversions.

Fig.1 Inverted stress fields under the fixed friction and variable friction.

a) Southern Zone

b) Central Zone

c) Northern Zone

d) Sürgü-Çardak

Figure S21. Illustration of the principal stresses and their uncertainties for the four zones plotted in lower-hemisphere stereonets.

Table 1. Focal mechanisms from [Güvercin et al., 2022], the same as the authors used for stress

ID	Event(yyyyymmdd hhmm)	Lat	Lon	Depth(km)	Strike	Dip	Rake	M _w	VR(%)	# Stations
7	200709150526	37.859	36.904	5.40	273	41	28	4.11	65	9
8	200709152328	37.868	36.889	6.09	249	35	0	3.94	71.9	9
60	201301080615	37.831	37.902	5.01	244	56	16	4.16	75.7	12
61	201301252124	37.896	37.946	6	265	44	46	3.25	75.5	9
64	201306162031	38.107	37.079	10	79	73	-8	4.10	76.1	20
75	201408081501	37.892	37.899	7.78	245	64	43	2.98	58.9	7
76	201501032324	37.902	37.876	4.85	255	40	0	3.37	80.5	12
83	201612160641	38.043	38.206	7.87	225	68	-45	3.24	71.9	8
92	201705161551	37.993	37.974	10.33	239	66	40	3.11	64.7	15
98	201712311138	37.827	37.754	5.60	255	66	-5	3.43	71.4	11
102	201804212341	37.855	37.793	5.52	55	45	61	3.21	72.3	19
106	201807141734	38.090	38.292	14	225	70	15	3.23	56.8	11
118	201909141347	38.136	37.771	7.58	62	50	-18	3.09	62.5	8
120	202001170006	38.050	38.089	8.53	274	85	-20	3.14	75.4	11

inversion in Fig.S21d.

2. Major concern on the geodetic deformation

I attached the Fig.S4 from the previous submission, in which the authors claimed they inferred rupture traces from their 3D deformations. In the recent response letter, they changed their statement as “We thank the reviewer for making us aware that there is a missing reference”. Since the fault trace was not derived from their 3D deformation and they didn’t include these 3D deformations in the finite fault inversion, what is the significance of the 3D deformation? It is an isolated part of the paper.

Figure S4. ALOS-2 LOS displacement (positive toward satellite) maps from the ascending track (a) and the descending track (b), respectively. The green lines show the surface rupture traces inferred by the SAR 3D deformation.

3. Major concern on 3D deformations and the input materials

The authors claimed they improved optical offsets through incorporating more Sentinel-2 images. But I didn't see any optical offset results in the main text and supplement. These important details on the input data for the 3D deformation inversion should not be ignored.

When inverting for the 3D deformations, a key accuracy indicator is the RMSE from the least square process. Please present the RMSE, which can partly show if there are unwrapping errors near the ruptured fault segments.

I am confused by Figs.3-4. Why the authors keep wrapping phase in a very large phase cycle. In the previous comment, I suggested to wrap them into a short cycle like $[-3\pi, 3\pi]$. But in this version, the authors just present them in $[-200, 200]$ radians. As I can see from the original interferograms, there is no signal in the near fault region due to too large deformations, the coherence of these regions should be near zero. How can the authors unwrap these signals that don't exist?

4. Major concern on rupture speed

Although the authors elaborated how they determine the rupture speed, I still can't tell how robust the rupture speed is constrained given the frequency range they are using to fit the strong motion waveform data. Some sensitivity tests will be helpful to understand.

Reviewer #4 (Remarks to the Author):

The authors have sufficiently addressed my suggestions and comments.

Reviewer #2 (Remarks to the Author):

The revised manuscript addresses part of my previous comments. But some of the critical points were still dodged by the authors.

Response: We appreciate the effort of the reviewer. See our point-by-point responses below.

1. Major concern on pre-stress heterogeneity from foreshocks

The number of foreshocks (55) that were used to derive the prestress orientation is too limited (Fig.S19), in particular when considering the size of the area (~300km x 300 km) and the complexity of the fault geometry. The way that the earthquakes are grouped is also questionable, for instance, in the southern segment, many of the events are quite far from the rupture (or plate boundary), it is not clear if these earthquakes can represent the stress condition of the M7.8 rupture; in the M7.6 event area (the yellow polygon), most of the events are actually located along the northern fault segment of the M7.8 earthquake, how can these events represent the stress field for both the M7.6 event and the northern segment of M7.8 event? The main conclusion of the paper is based on the stress orientation derived from these 55 earthquakes, it is critical to show that the stress field they obtained robustly represent the stress field that governed the rupture of the doublet.

Response: As we mention in the revised manuscript at lines 198-199, the focal mechanisms (shown in Figure S21 reproduced below) clearly suggest a gradual variation of the orientation of the principal stresses. The polygons were defined to quantify these variations with the requirement to have appropriate data in each polygon (we have shown that the number and diversity of focal mechanisms are sufficient as they meet the 30° RMS diversity criteria defined by Hardebeck & Hausskon [2001], see Fig. S22). We acknowledge that there is no objective way to choose these polygons, which is common practice, but we are confident that the rotation of the principal stress direction is required and not much sensitive to the choice of these polygons.

Figure S21

I notice there are other recent publications on the stress field, it would be helpful that the authors compare their results with those derived from other methods.

Response: In addition, we note that other studies using focal mechanisms and/or the strain rate from the geodetic velocity field have also found the same spatial rotation of stresses along the rupture that we find (Gabriel et al., 2023; Xu et al., 2023). This is clearly shown by the figures below from the studies.

[REDACTED]

Figure above from Gabriel et al. (2023): “zoomed view of East Anatolian fault zone principal strain rate directions in purple (first component) and pink (second component) from Weiss et al. (2020). In dark and light gray, we show the seismologically inferred maximum and minimum principal horizontal stress components from Güvercin et al. (2022), as well as in dark and light blue, the maximum and minimum principal horizontal stress orientations used in this study.”

[REDACTED]

Figure above from Xu et al. (2023): “The red bars indicate the optimal slip orientation computed based on the regional strain rate field by assuming a rock friction coefficient of 0.6 and the local strain direction being the same as the regional horizontal strain rate direction.”

Xu, L., Mohanna, S., Meng, L., Ji, C., Ampuero, J.P., Yunjun, Z., Hasnain, M., Chu, R. and Liang, C., 2023. The overall-subshear and multi-segment rupture of the 2023 Mw7.8 Kahramanmaraş, Turkey earthquake in millennia supercycle. *Communications Earth & Environment*, 4(1), p.379.

Gabriel, A.A., Ulrich, T., Marchandon, M., Biemiller, J. and Rekoske, J., 2023. 3D Dynamic Rupture Modeling of the 6 February 2023, Kahramanmaraş, Turkey M w 7.8 and 7.7 Earthquake Doublet Using Early Observations. *The Seismic Record*, 3(4), pp.342-356.

I also have questions on the stress inversion results (Fig.S21). Take the Surgu-Cardak region (the yellow polygon in Fig.S19) as an example, I use the same 14 focal mechanisms from [Güvercin et al., 2022], the same source as the authors obtained the focal mechanism, and perform similar stress inversion with uncertainty estimated by a bootstrapping method (similar as the authors did). If I set the friction to 0.6, again the same as the authors did, the resulted stress orientation is shown in upper panels of Fig.1, where the uncertainty of sigma-3 is much larger than that of sigma-1 and sigma-2. Even for sigma-1 and sigma-2, the uncertainty is much larger than that shown in Fig.S21d, no mention that if one allows a variable friction coefficient on the fault (lower panels in Fig.1), the uncertainty of stress inversion is getting even larger. With such strong contrasts and small number of foreshocks, I don't understand how the authors can obtain such tight uncertainties in their inversions.

Fig.1 Inverted stress fields under the fixed friction and variable friction.

Figure S21. Illustration of the principal stresses and their uncertainties for the four zones plotted in lower-hemisphere stereonets.

Table 1. Focal mechanisms from [Güvercin et al., 2022], the same as the authors used for stress

ID	Event(yyyyymmdd hhmm)	Lat	Lon	Depth(k m)	Strike	Dip	Rake	M _w	VR(%)	# Stat ions
7	200709150526	37.859	36.904	5.40	273	41	28	4.11	65	9
8	200709152328	37.868	36.889	6.09	249	35	0	3.94	71.9	9
60	201301080615	37.831	37.902	5.01	244	56	16	4.16	75.7	12
61	201301252124	37.896	37.946	6	265	44	46	3.25	75.5	9
64	201306162031	38.107	37.079	10	79	73	-8	4.10	76.1	20
75	201408081501	37.892	37.899	7.78	245	64	43	2.98	58.9	7
76	201501032324	37.902	37.876	4.85	255	40	0	3.37	80.5	12
83	201612160641	38.043	38.206	7.87	225	68	-45	3.24	71.9	8
92	201705161551	37.993	37.974	10.33	239	66	40	3.11	64.7	15
98	201712311138	37.827	37.754	5.60	255	66	-5	3.43	71.4	11
102	201804212341	37.855	37.793	5.52	55	45	61	3.21	72.3	19
106	201807141734	38.090	38.292	14	225	70	15	3.23	56.8	11
118	201909141347	38.136	37.771	7.58	62	50	-18	3.09	62.5	8
120	202001170006	38.050	38.089	8.53	274	85	-20	3.14	75.4	11

inversion in Fig.S21d.

Response: We note it is reassuring that the reviewer obtained a MAP stress tensor very close to the one we obtained for the Surgu-cardak zone. The reviewer however claimed that the uncertainty is much larger than what we obtained. But we note that the results reported by reviewer do not show the uncertainties; it shows the position of all bootstrap model solutions from the inversion. In the figure we show in the supplements we plot the probability density, not the position or cloud of all model solutions. The latter is problematic because with an infinite number of bootstrap simulations the entire stereogram would be filled with a cloud of points with no ability to determine the region of higher or lower model uncertainty. Thus, instead we have chosen to plot the probability density of the solutions which is able to show the varying density regions of uncertainty. We note that to provide these plots we do not plot bins that contain less than 20 model solutions to show regions with a stable result, but this is still consistent with the color scale now included in the figure. Therefore, comparing a cloud of all bootstrap model solutions to a probability density gives a false impression that the confidence regions are artificially smaller, and we suspect that the density of points in the two inversions are actually very similar. We have now clarified in the supplementary figure caption in Figure S21, which shows the density of points which is an information that was likely missed as we mistakenly did not include a figure legend that explained the meaning of the colored regions.

Fig. S23 Illustration of the principal stresses and their uncertainties for the four zones plotted in lower-hemisphere stereonets from 4000 bootstrap simulations. The colored regions show the probability density; we do not include bins that contain less than 20 model solutions from the bootstrap in order to show regions of stable solutions. Red triangle shows the direction of SHmax (Lund & Townend, 2007).

Lund, B. and Townend, J., 2007. Calculating horizontal stress orientations with full or partial knowledge of the tectonic stress tensor. *Geophysical Journal International*, 170(3), pp.1328-1335.

2. Major concern on the geodetic deformation

I attached the Fig.S4 from the previous submission, in which the authors claimed they inferred rupture traces from their 3D deformations. In the recent response letter, they changed their statement as “We thank the reviewer for making us aware that there is a missing reference”. Since the fault trace was not derived from their 3D deformation and they didn’t include these 3D deformations in the finite fault inversion, what is the significance of the 3D deformation? It is an isolated part of the paper.

Figure S4. ALOS-2 LOS displacement (positive toward satellite) maps from the ascending track (a) and the descending track (b), respectively. The green lines show the surface rupture traces inferred by the SAR 3D deformation.

Response: The calculated 3D surface displacements are highly consistent with the measurements made from the optical images as shown in Figure 2a. As explained at lines 89-90, the 3D near-fault deformation measurements derived from the optical images were used to constrain the fault geometry and the amount of slip at shallow depth ($z < 2.5\text{km}$). As you pointed out in the previous comments, the 3D deformation field contains co-seismic deformations from two earthquakes, making it difficult to precisely separate the individual contributions of each earthquake. Therefore, we did not use the entire 3D deformation field data to constrain the slip distribution of the two earthquakes in the finite fault inversion.

3. Major concern on 3D deformations and the input materials

The authors claimed they improved optical offsets through incorporating more Sentinel-2 images. But I didn't see any optical offset results in the main text and supplement.

Response: The optical offsets are shown in Figure 2. They were used to help inform our initial understanding of the location, geometry and kinematics of the faults that slipped most during these earthquakes (e.g., whether there was large vertical slip or not). They also allow to measure surface slip with account for distributed inelastic shear that generally is not measurable from field survey of seismic ruptures. We explain how these data are used at lines 62, 69, 89, 90.

Figure above shows the optical (Sentinel-2) and radar (Sentinel-1) pixel offset data used to invert for the 3D surface deformation. The black arrows show the direction of motion measured from the pixel offset, and red shows the orbit direction (not shown for optical). We have now included it in the supplement as Figure S1.

These important details on the input data for the 3D deformation inversion should not be ignored. When inverting for the 3D deformations, a key accuracy indicator is the RMSE from the least square process. Please present the RMSE, which can partly show if there are unwrapping errors near the ruptured fault segments.

Response: We have now shown the RMSE misfit to all the pixel offset data used (note we do not use the unwrapped phase, only pixel offsets). Below shows the RMSE which we have now included in the supplements. The RMSE is spatially variable and is largely due to larger noise in regions of higher relief. Median RMSE for the entire region is 0.40 m (shown by red line).

I am confused by Figs.3-4. Why the authors keep wrapping phase in a very large phase cycle. In the previous comment, I suggested to wrap them into a short cycle like $[-3\pi, 3\pi]$. But in this version, the authors just present them in $[-200, 200]$ radians. As I can see from the original interferograms, there is no signal in the near fault region due to too large deformations, the coherence of these regions should be near zero. How can the authors unwrap these signals that don't exist?

Response: As discussed in an earlier round of revision, we did wrap the interferograms with a much shorter cycle. Below are the wrapped results we presented in that response. We do not see any significant phase unwrapping errors from these results. We didn't include the image with a short cycle wrapping as it would not be legible in print.

Ascending track 184, 220905-230220

Descending track d77, 220916-230217

As we explain at lines 397-398 of the revised manuscript, before unwrapping, we visually inspected the original wrapped interferograms, and masked out the areas near the faults where the fringes are too dense to unwrap. These masked areas are very clear in the rewrapped interferogram shown above (Figure S6 of the revised manuscript). Note that the original wrapped interferograms are very large and can show many clear fringes not far from the fault. These fringes may not be clearly seen from the wrapped interferograms shown above (left panels). The original wrapped interferograms are too large and we cannot put them in this response letter.

4. Major concern on rupture speed

Although the authors elaborated how they determine the rupture speed, I still can't tell how robust the rupture speed is constrained given the frequency range they are using to fit the strong motion waveform data. Some sensitivity tests will be helpful to understand.

Response: As explained at line 458 of the revised manuscript, we determine the optimal rupture velocity based on how the data fit varies a function of the rupture velocities using a grid search (see Figures S12 and S16). We chose the frequency band as done commonly in kinematic inversions. The Earth's interior exhibits significant heterogeneity and anisotropy, resulting in complex and variable wave propagation paths and velocities at high frequencies which are generally not resolvable with the 1-D layered velocity model as used in our study. The manuscript specifies at line 378 the frequency range used in our inversions.

Reviewer #4 (Remarks to the Author):

The authors have sufficiently addressed my suggestions and comments.

Response: Thank you for your positive feedback and for taking the time to review our manuscript.